# DESCRIBE-TO-SCORE: TEXT-GUIDED EFFICIENT IMAGE COMPLEXITY ASSESSMENT

## ABSTRACT

Accurately assessing image complexity (IC) is essential for many vision tasks, yet existing approaches rely almost exclusively on visual features and therefore fail to capture the high-level semantics that humans often use when judging complexity. We introduce a multimodal perspective for IC modeling by integrating visual representations with caption-derived textual semantics. This integration enriches the representational space and provides complementary structural cues that are difficult to infer from vision alone. From an information theoretic and representation viewpoint, we offer an idealized analysis suggesting how semantic guidance can regularize the hypothesis space and support more stable generalization. We present D2S (Describe-to-Score), a framework that generates natural-language descriptions using a pretrained vision–language model and aligns the visual encoder with textual structure through feature alignment and entropy distribution alignment. These mechanisms encourage the visual backbone to internalize semantic regularities during training. Importantly, D2S employs text only during training and maintains a vision-only inference pipeline with no additional computational overhead. Experiments show that D2S achieves state-of-the-art performance on the IC9600 benchmark and remains competitive on no-reference image quality assessment (NR-IQA) tasks. Additional studies demonstrate robustness across captioners and prompt designs, and improved semantic transferability on downstream probing tasks, highlighting the effectiveness of multimodal guidance for complexity-related modeling.

## 1 INTRODUCTION

Image complexity (Forsythe, 2009) (IC) is a fundamental factor in human visual perception, influencing aesthetic judgment and memorability (Singh & Shukla, 2017). In computer vision, accurate image complexity assessment (ICA) supports automatic annotation, active learning, and hard example mining by identifying informative samples and improving learning efficiency and generalization (Feng et al., 2022). Early approaches relied on statistical measures such as fractal dimension, entropy (Li et al., 2021), and edge density (Dai et al., 2022) to characterize structural richness. However, these handcrafted indicators suffer from subjective criteria, inconsistent definitions, and limited cross-domain robustness. With the development of deep learning, convolutional neural networks (CNNs) and vision transformers (ViTs) (Liu et al., 2025a) have significantly improved prediction accuracy by learning hierarchical visual features. Nevertheless, current models still primarily emphasize low-level cues such as texture and color, while lacking explicit mechanisms to capture high-level semantics including object count, category, and spatial organization. In addition, many visual-only models provide limited interpretability (Chen et al., 2015; Shen et al., 2024).

Recent work has begun incorporating semantic signals, such as object counts (Shen et al., 2024) or motion patterns (Li et al., 2025), into complexity modeling and has reported promising improvements. Meanwhile, we note that human perception of complexity depends on an integrated understanding of both local visual attributes and high-level semantics, including the number, types, and relationships of objects as well as implied events. This observation motivates the following central question:

*How can image complexity be computationally evaluated in a human-like manner that integrates low-level visual information with high-level semantic cues?*

To explore this question, we first examined whether captions contain useful IC-related information. We used a vision–language model (VLM) (Li et al., 2022) to generate captions for images and trained a model to predict IC based solely on textual descriptions. Surprisingly, caption-only prediction already achieved a PCC of 0.8251, as illustrated in Figure 1 (right) and detailed in Table 11. This result suggests that captions indeed encode meaningful semantic attributes correlated with perceived complexity. Building on this observation, we integrate the visual and textual branches in Figure 1 and introduce D2S (**D**escribe-to-**S**core), which leverages caption-derived semantics to guide visual complexity modeling.

The core idea of D2S is ***Describe first; then Score.*** **(1) Describing.** Captions generated by a pretrained VLM capture object concepts, relations, and structural semantics. **(2) Scoring.** Through vision–text alignment, semantic information shapes the visual representation space, enabling the model to utilize both high-level semantics and low-level cues for more reliable complexity assessment. On IC9600 (Feng et al., 2022), text-guided prediction improves PCC by 0.0146 over the visual-only baseline (Figure 1, middle).

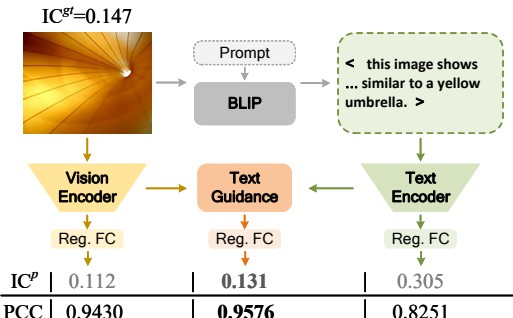

Figure 1: Text-guided image complexity assessment. $IC^{gt}$ and $IC^{p}$ denote the ground-truth and predicted IC scores, respectively.

We further examine the rationale behind multimodal fusion (Park & Kim, 2023). From an information-theoretic viewpoint, entropy analysis (Yang & Nataliani, 2017) shows that fused visual–text features exhibit higher entropy than vision-only features, providing a richer approximation of true complexity distributions (Section 2.2). From a generalization-theoretic perspective, Empirical Rademacher Complexity (Mohri & Rostamizadeh, 2008) reveals that semantic signals can regularize the visual hypothesis space and effectively reduce the degrees of freedom in the learned representation, tightening generalization bounds (Section 2.3). Guided by these insights, we introduce entropy distribution alignment (EAL) and feature alignment (FAL) to improve cross-modal consistency. Extensive experiments validate their effectiveness: D2S achieves state-of-the-art (SOTA) performance on IC9600 (Figure 2) with significantly reduced inference latency (Figure 3). Moreover, D2S generalizes strongly to no-reference image quality assessment (NR-IQA) (Mittal et al., 2012), achieving leading results on KADID-10k (Lin et al., 2019). Our main contributions are as follows:

• We propose D2S, a vision–text fusion framework in which semantic cues enrich visual representations and regularize the hypothesis space, improving both accuracy and generalization.

• We develop entropy distribution alignment and feature alignment to bridge the visual–text gap, enhancing cross-modal consistency and robustness.

• Extensive experiments on IC9600, KADID-10K, and related benchmarks establish SOTA performance with fast inference, demonstrating both efficiency and cross-task adaptability.

## 2 PRELIMINARY

### 2.1 TASK DEFINITION

Given an input image $I$ from the dataset $\mathcal{D}$, its ground-truth complexity label is denoted as $y \in (0, 1)$. An arbitrary caption generation model $g_\phi$ produces a text description $S = g_\phi(I)$. We then train a VLM $f_\theta$, which consists of a visual encoder and a text encoder, to predict the IC score $\hat{y} = f_\theta(I, S)$. The learning objective is to minimize the prediction error:

$$\min_\theta \mathbb{E}_{(I,y)\sim\mathcal{D}}\big[\ell\big(f_\theta(I, S), y\big)\big] \tag{1}$$

where $\ell$ denotes the loss function (e.g., MSE), and $\theta$ represents the model parameters. For the single-modal case using only the image input, the prediction result is written as $\hat{y}_v = f_\theta(I)$. In what

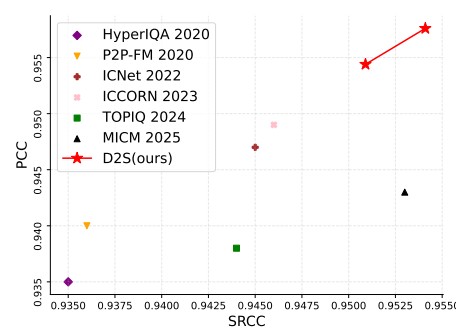

Figure 2: Accuracy comparison with SOTA (higher is better; upper right is preferred).

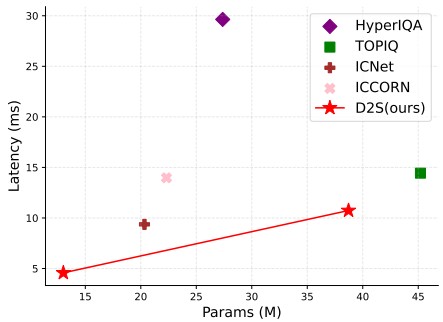

Figure 3: Efficiency comparison on RTX 3090. (lower is better; lower left is preferred).

follows, we focus on understanding how multimodal training reshapes the representation and why semantic information can help complexity assessment under suitable conditions. We do not assume a priori that $\hat{y}$ always dominates $\hat{y}_v$. Instead, we analyze the mechanisms through which multimodal fusion can provide advantages over the visual-only baseline.

## 2.2 ENTROPY AND COMPLEXITY

**Proposition 1.** *Image complexity tends to increase with the growth of visual diversity and semantic diversity. Let the original entropy be $H(I)$, the entropy of visual features be $H_v^F(I)$, and the entropy of semantic features be $H_s^F(S)$. We model the fused entropy as*

$$H^F(I) = \alpha\, H_v^F(I) + \beta\, H_s^F(S) \tag{2}$$

*where $\alpha, \beta > 0$ are weighting coefficients.* This linear relation is introduced as a tractable approximation to capture the joint contribution of visual and semantic cues, rather than as a universal identity for all multimodal systems. We empirically verify in Appendix A.1 that, for our trained model, there exist positive coefficients $\alpha$ and $\beta$ such that $H^F(I)$ closely matches the entropy measured from fused features.

**Implications of Proposition 1.** Semantic information complements visual information and enriches representation diversity by increasing entropy, so that the fused features can encode structural cues that are more closely aligned with perceived complexity. Consequently, in complexity assessment tasks, multimodal models are expected to benefit from a more expressive feature space than their single-modal counterparts. Although $H^F(I) > H_v^F(I)$ does not hold universally for all choices of $\alpha$ and $\beta$, empirical analysis shows that the fused entropy tends to exceed visual entropy for the vast majority of samples in our trained model.

## 2.3 GENERALIZATION VIA RADEMACHER COMPLEXITY

**Definition 1 (Empirical Rademacher Complexity** (Mohri & Rostamizadeh, 2008)). *Suppose the hypothesis space is $\mathcal{F}$ with sample size $n$. If the feature representation $\phi(I, S)$ satisfies the boundedness condition $|\phi(I, S)| \leq B$, and its feature dimension is $d$, then the empirical Rademacher complexity satisfies*

$$\widehat{\mathcal{R}}_S(\mathcal{F}) \leq \frac{B\sqrt{d}}{\sqrt{n}}. \tag{3}$$

**Lemma 1 (Dimensionality Reduction via Semantics).** *Let the visual and semantic features be denoted as $X_v$ and $X_s$, respectively. If the semantic features $X_s$ exert a compressing or regularizing effect on the visual features $X_v$, the effective dimension $d$ will be reduced to $d'$, with $d' < d$. In this case,*

$$\widehat{\mathcal{R}'}_S(\mathcal{F}) \leq \frac{B\sqrt{d'}}{\sqrt{n}} \leq \frac{B\sqrt{d}}{\sqrt{n}}. \tag{4}$$

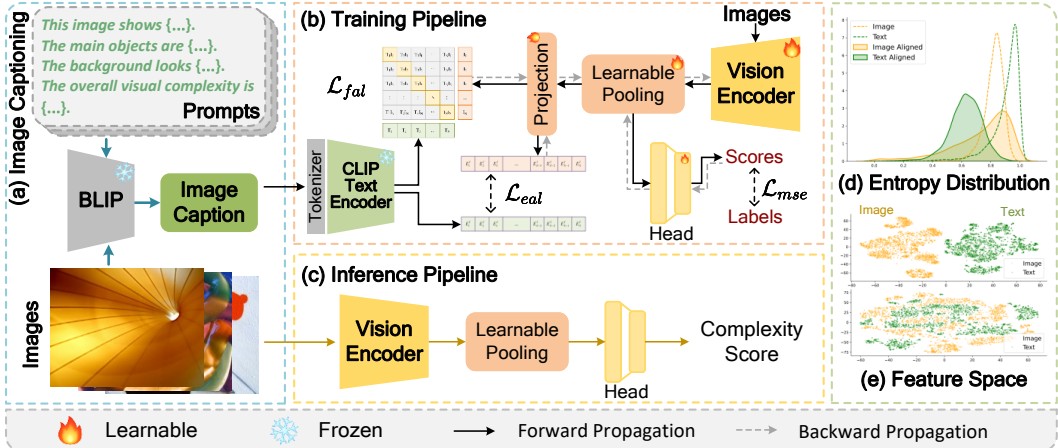

Figure 4: **Overall Architecture of the D2S framework.** (a) BLIP with a fill-in prompt generates captions. (b) Vision and text encoders extract features for regression loss $\mathcal{L}_{\mathrm{mse}}$ and the alignment losses $\mathcal{L}_{\mathrm{fal}}$, $\mathcal{L}_{\mathrm{eal}}$. (c) At inference, only image input is used for score prediction. (d) Entropy distribution before and after alignment (Dotted line: before alignment; solid line: after alignment).(e) Feature space before (top) and after (bottom) feature alignment.

**Theorem 2 (Generalization Enhancement).** *Under the framework of visual–text fusion, if semantic inputs reduce the effective feature dimension in the sense of Lemma 1, then the empirical Rademacher complexity decreases, thereby enhancing the generalization capability of the model.*

**Proof.** *From* Lemma 1, *the effective dimension is reduced from* $d$ *to* $d'$. *Since the upper bound of* $\widehat{\mathcal{R}}_S(\mathcal{F})$ *is proportional to* $\frac{B\sqrt{d'}}{\sqrt{n}}$, *a smaller* $d'$ *leads to a tighter bound. According to statistical learning theory, a lower Rademacher complexity implies a smaller generalization error bound. Therefore, when semantic information effectively regularizes the representation, visual–text fusion can improve model generalization.*

**Implications of Proposition 1 and Theorem 2.** The advantages of visual–text fusion can be summarized in two aspects. (1) It increases representational diversity by integrating semantic cues, thereby providing a richer approximation to the true complexity of images. (2) It can reduce the effective hypothesis space dimension when semantic signals act as a regularizer on visual features, which in turn leads to tighter generalization bounds. These results describe conditions under which multimodal methods are expected to outperform single-modal baselines in complexity assessment and related tasks, and they are consistent with the empirical improvements observed in our experiments.

## 3 METHOD

### 3.1 DESCRIBE-TO-SCORE

Our proposed D2S framework aims to jointly model low-level visual features and high-level semantic information through text guidance, thereby achieving robust ICA. Figure 4 shows the overall workflow. Given an input image, a pre-trained BLIP first generates captions. The visual encoder then extracts visual features, while the text encoder extracts textual features. The vision-text alignment module of D2S consists of entropy distribution alignment and feature alignment. During training, textual information acts as a semantic teacher that reshapes the visual representation. However, **it is not directly used in the computation of the final complexity score**. The complexity score is produced solely from the aligned visual features through an MLP head. This design ensures that D2S benefits from semantic guidance during learning while remaining vision-only at inference.

**Image captioning.** We adopt a well-designed fill-in-the-blank prompt template fed into BLIP-Large (Li et al., 2022) to generate captions with images. We design four sentences (Figure 5)

to obtain complementary descriptions focusing on **Simple Description**, **Object Categories**, **Background**, and **Global Visual Complexity**, respectively.

For each image in the training set, the prompts are applied sequentially, and the outputs of BLIP are combined to form a complete caption. Each image and its caption constitute an image–text pair, and captions are not re-generated during training. Using fixed captions stabilizes supervision and avoids injecting randomness into the semantic branch. We provide examples in Appendix A.8.

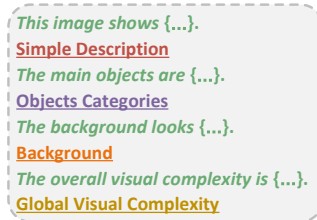

Figure 5: Prompt template.

**Vision encoder & Text encoder.** The vision encoder in D2S is based on ResNet (He et al., 2016), but the original global pooling and fully connected layers are replaced with a learnable pooling layer that outputs a one-dimensional feature vector for each image. We employ the pre-trained CLIP (Radford et al., 2021) text encoder as the text encoder in D2S, and use the CLIP tokenizer to segment each caption. The text encoder remains **frozen during training**, while gradients are updated only for the other components. Freezing the text encoder prevents semantic drift and maintains a stable semantic reference space.

**Vision-Text Connector.** Many prior works employ an MLP-style projection layer (Chen et al., 2020) to map visual and textual features into the same dimensional space. Following this practice, we apply **one linear layer** as a projection for the vision encoder, aligning it with the dimensionality of the text features. This design ensures that text features can effectively guide visual features within a shared representation space. **Note, the projection will be discarded during inference.** Thus, the connector serves purely as a training-time alignment mechanism with no impact on model deployment.

### 3.2 ENTROPY DISTRIBUTION ALIGNMENT

In the theoretical analysis, we pointed out that the weighted fusion of visual feature entropy $H_v^F(I)$ and semantic feature entropy $H_s^F(S)$ can better capture image complexity. However, during model learning, we observe that the empirical distributions of the two modalities show a noticeable bias (Figure 4(d), dotted line). Such bias may introduce additional uncertainty in the fusion stage and weaken the consistency of cross-modal complexity measurement. To address this issue, we propose entropy distribution alignment. By aligning the entropy distributions, D2S stabilizes the multimodal supervision signal and encourages a more coherent complexity-aware representation.

**Computing feature entropy.** To compute the entropy of visual and textual features, we follow the standard practice in information-theoretic analysis of neural representations. Given an intermediate feature vector $z \in \mathbb{R}^d$, we first normalize it using a temperature-scaled softmax to obtain a discrete probability distribution:

$$p_k = \frac{\exp(z_k/\tau)}{\sum_{j=1}^d \exp(z_j/\tau)}, \qquad k = 1, \ldots, d, \tag{5}$$

where $\tau$ is a temperature parameter that controls the smoothness of the distribution (*we set $\tau = 0.07$ unless otherwise specified*). The feature entropy is then computed using the Shannon entropy:

$$H(z) = -\sum_{k=1}^d p_k \log p_k. \tag{6}$$

For the visual and textual branches, we compute $H_v^F(I)$ and $H_s^F(S)$ by applying the above procedure to the outputs of the vision encoder and the text encoder, respectively. To avoid numerical instability, we add a small constant $\epsilon = 10^{-8}$ inside the logarithm. This entropy formulation treats the feature magnitude pattern as an empirical distribution over channels and provides a simple, consistent proxy for representational diversity.

**Entropy Buffer.** We establish and maintain two FIFO entropy buffers, $B_v$ and $B_s$, each with capacity $M$ to store visual and textual feature entropy. Before training starts, we duplicate D2S and freeze it, referring to this frozen copy as the Momentum Model (MoM). At each iteration, we first

use the MoM in inference mode to obtain the image and text features of all samples in the mini-batch, and then compute their entropy using Eq.(14). The visual and textual entropy are stored in their corresponding buffers. Once a buffer reaches half of its capacity, we begin to compute the entropy distribution alignment loss. We update the buffers by adding the new mini-batch samples and removing the oldest entries with the same number, thereby reducing distributional bias between old and new samples. Furthermore, we update the MoM at each iteration using an exponential moving average (EMA) of gradients (He et al., 2020), and employ the updated MoM to refresh buffer entries with a *refresh step*. This mechanism keeps the entropy statistics temporally consistent and mitigates drift between modalities. The full algorithm of EAL is provided in Appendix A.5.

**Entropy Distribution Alignment Loss** $\mathcal{L}_{\mathrm{eal}}$. We define the alignment loss using the energy distance (Székely & Rizzo, 2013). Let

$$V = \{v_i\}_{i=1}^M, \qquad S = \{s_j\}_{j=1}^M \tag{7}$$

denote the entropy values stored in $B_v$ and $B_s$. The energy distance is defined as

$$D_E^2(V, S) = 2\mathbb{E}[\|V - S\|] - \mathbb{E}[\|V - V'\|] - \mathbb{E}[\|S - S'\|], \tag{8}$$

where $V, V'$ are i.i.d. samples from $B_v$ and $S, S'$ are i.i.d. samples from $B_s$. The empirical estimate is

$$\mathcal{L}_{\mathrm{eal}} = 2\frac{1}{M^2} \sum_{i=1}^M \sum_{j=1}^M |v_i - s_j| - \frac{1}{M(M-1)} \Big( \sum_{i \neq i'} |v_i - v_{i'}| - \sum_{j \neq j'} |s_j - s_{j'}| \Big). \tag{9}$$

Minimizing this loss encourages the two modalities to share similar entropy statistics, improving their consistency during training.

### 3.3 Feature Alignment

To further exploit textual information, we align visual and textual features in a shared representation space. Inspired by CLIP (Radford et al., 2021), we introduce a contrastive loss that enforces correspondence between visual and textual representations. This encourages the visual encoder to absorb structural regularities present in the semantic space, complementing the entropy-level alignment of EAL. Let the outputs of the visual and text encoders be

$$z_v = f_v(I), \qquad z_s = f_s(S), \tag{10}$$

which are then projected into the joint space through

$$\tilde{z}_v = W_v z_v, \qquad \tilde{z}_s = W_s z_s, \tag{11}$$

where $W_v$ and $W_s$ denote projection parameters, and $W_s = I$ in this work. With temperature parameter $\tau$, the InfoNCE loss (van den Oord et al., 2019) is:

$$\mathcal{L}_{\mathrm{fal}} = -\frac{1}{N} \sum_{i=1}^N \log \frac{\exp(\mathrm{sim}(\tilde{z}_v^i, \tilde{z}_s^i)/\tau)}{\sum_{j=1}^N \exp(\mathrm{sim}(\tilde{z}_v^i, \tilde{z}_s^j)/\tau)}. \tag{12}$$

This loss serves as a semantic regularizer that improves representation geometry and reduces redundancy, helping to control the effective dimensionality of the visual manifold. The regression head is optimized using MSE $\mathcal{L}_{\mathrm{mse}}$. The final objective is

$$\mathcal{L} = \mathcal{L}_{\mathrm{mse}} + \lambda \mathcal{L}_{\mathrm{eal}} + \gamma \mathcal{L}_{\mathrm{fal}}, \tag{13}$$

where $\lambda$ and $\gamma$ control the relative weights of $\mathcal{L}_{\mathrm{eal}}$ and $\mathcal{L}_{\mathrm{fal}}$.

## 4 Experiments

The reported results are averaged over three random seeds (42, 826, 1215), and we additionally report the standard deviation (std) in Table 1. More training details are provided in Appendix A.2.

Table 1: **Comparison with state-of-the-art image complexity assessment methods on IC9600.** Entropy buffer size is 2048 with a refresh step of 50 ($\sim$ 40 entropy were refreshed per iteration) and momentum 0.995. $\lambda$ and $\gamma$ are set to 5 and 0.01, respectively. $\dagger$ denotes an unsupervised method. The results of ICNet and ICCORN are our re-implementation. The **bolded** portion marks the optimal outcome. The underlined portion denotes the second-best. SRCC and PCC, larger $\uparrow$ is better. RMSE, RMAE, Params (M: million, B: billion), and Latency (ms: millisecond, s: second), smaller $\downarrow$ is better.

| Method | SRCC | PCC | RMSE | RMAE | Params | Latency |
|---|---|---|---|---|---|---|
| MoCo$^\dagger$ (He et al., 2020) | 0.759 | 0.748 | - | - | - | - |
| SAE$^\dagger$ (Saraee et al., 2020b) | 0.865 | 0.860 | 0.074 | 0.240 | - | - |
| CLIC$^\dagger$ (Liu et al., 2024) | 0.866 | 0.858 | - | - | - | - |
| CLICv2$^\dagger$ (Liu et al., 2025a) | 0.879 | 0.870 | - | - | - | - |
| HyperIQA (Su et al., 2020) | 0.935 | 0.935 | 0.067 | 0.229 | 27.38M | 29.638ms |
| P2P-FM (Ying et al., 2020) | 0.940 | 0.936 | 0.056 | 0.208 | - | - |
| TOPIQ (Chen et al., 2024) | 0.938 | 0.944 | 0.049 | - | 45.20M | 14.434ms |
| ICNet (Feng et al., 2022) | 0.9446 | 0.9470 | 0.0582 | 0.2156 | 20.33M | 9.369ms |
| std | ±.0029 | ±.0011 | ±.0021 | ±.0051 | | |
| ICCORN (Guo et al., 2023) | 0.9455 | 0.9490 | 0.0526 | 0.2085 | 22.31M | 13.973ms |
| std | ±.0011 | ±.0018 | ±.0013 | ±.0053 | | |
| MICM (Li et al., 2025) | 0.943 | 0.953 | 0.060 | - | $\sim$11B | $\sim$180s |
| D2S-R18 (ours) | 0.9509 | 0.9544 | **0.0495** | **0.1962** | **13.02M** | **4.573ms** |
| std | ±.0026 | ±.0009 | ±.0016 | ±.0050 | | |
| D2S-R50 (ours) | **0.9541** | **0.9576** | 0.0496 | 0.1963 | 38.72M | 10.738ms |
| std | ±.0004 | ±.0010 | ±.0028 | ±.0062 | | |

Table 2: **Performance of small samples training on IC9600.** SST@X denotes few-shot training with X samples. Epoch: 5. Hyper-parameters are same as Table 1. SRCC and PCC, larger $\uparrow$ is better. RMSE and RMAE, smaller $\downarrow$ is better.

| Method | SRCC | PCC | RMSE | RMAE | SRCC | PCC | RMSE | RMAE |
|---|---|---|---|---|---|---|---|---|
| | SST@10 | | | | SST@50 | | | |
| ICNet | 0.5178 | 0.5057 | 0.1527 | 0.3429 | 0.5644 | 0.5864 | 0.1514 | 0.3422 |
| ICCORN | 0.5221 | 0.5102 | 0.1497 | 0.3393 | 0.5704 | 0.5914 | 0.1484 | 0.3386 |
| D2S-R18 | 0.6664 | 0.6839 | 0.1302 | 0.3199 | 0.8174 | 0.8332 | **0.1030** | **0.2858** |
| D2S-R50 | **0.6864** | **0.6972** | **0.1190** | **0.3055** | **0.8515** | **0.8528** | 0.1133 | 0.3025 |
| | SST@100 | | | | SST@500 | | | |
| ICNet | 0.8239 | 0.8259 | 0.1668 | 0.3820 | 0.8942 | 0.9054 | 0.0830 | 0.2582 |
| ICCORN | 0.8276 | 0.8294 | 0.1605 | 0.3747 | 0.8977 | 0.9080 | 0.0791 | 0.2527 |
| D2S-R18 | 0.8552 | 0.8547 | **0.0964** | **0.2787** | 0.9115 | 0.9182 | **0.0707** | **0.2326** |
| D2S-R50 | **0.8680** | **0.8731** | 0.1092 | 0.2913 | **0.9182** | **0.9246** | 0.0709 | 0.2349 |

## 4.1 Main Results

**Comparison with State-of-the-Art Methods.** We benchmark the proposed D2S against both unsupervised and supervised ICA approaches on IC9600, with results summarized in Table 1. Unsupervised methods exhibit limited performance (best SRCC 0.879), while supervised variants benefit from labeled data. MICM achieves a strong PCC of 0.953 but relies on extremely large models ($\sim$ 11B parameters, $\sim$ 180s latency), making it impractical for most applications.

By contrast, D2S consistently surpasses all baselines in accuracy, efficiency, and stability. D2S-R18 achieves an SRCC of 0.9509 with only 13.02M parameters, reducing latency to 4.573ms and outperforming ICCORN at one-fourth of its computational cost. Scaling to ResNet50 further boosts

Table 3: **Cross-dataset generalization of D2S on ICA datasets.** Epoch: 5. Hyper-parameters are same as Table 1. We also provided the training results based on the full data in Appendix A.6.1 with more methods. SRCC and PCC, larger ↑ is better. RMSE and RMAE, smaller ↓ is better.

| Method | SRCC | PCC | RMSE | RMAE | SRCC | PCC | RMSE | RMAE |
|---|---|---|---|---|---|---|---|---|
| | Nagle4k (Nagle & Lavie, 2020b) | | | | VISC-C (Kyle-Davidson et al., 2023) | | | |
| ICNet | 0.7851 | 0.7666 | 0.1082 | 0.2917 | 0.7219 | 0.7111 | 0.1415 | 0.3367 |
| ICCORN | 0.7905 | 0.7706 | **0.1070** | **0.2891** | 0.7251 | 0.7155 | 0.1407 | 0.3340 |
| D2S-R18 | 0.7976 | 0.7748 | 0.1102 | 0.2949 | 0.7291 | 0.7165 | **0.1399** | **0.3350** |
| D2S-R50 | **0.7976** | **0.7765** | 0.1126 | 0.2989 | **0.7317** | **0.7170** | 0.1421 | 0.3375 |
| | Savoias (Saraee et al., 2020b) | | | | VISC-CI (Kyle-Davidson et al., 2023) | | | |
| ICNet | 0.6813 | 0.6793 | 0.1721 | 0.3736 | 0.6802 | 0.6876 | **0.1549** | **0.3571** |
| ICCORN | 0.6835 | 0.6811 | 0.1719 | 0.3720 | 0.6825 | 0.6924 | 0.1563 | 0.3591 |
| D2S-R18 | 0.6780 | 0.6825 | 0.1706 | 0.3717 | **0.6853** | **0.6982** | 0.1580 | 0.3617 |
| D2S-R50 | **0.6845** | **0.6882** | **0.1700** | **0.3713** | 0.6828 | 0.6963 | 0.1622 | 0.3660 |

Table 4: **Cross-task transfer of D2S to NR-IQA.** D2S were trained for 5 epochs with input resolution 384. The method references and implementation details are provided in the Appendix A.2.

| Method | Source | KADID-10K | | KonIQ-10K | | TID2013 | |
|---|---|---|---|---|---|---|---|
| | | SRCC ↑ | PCC ↑ | SRCC ↑ | PCC ↑ | SRCC ↑ | PCC ↑ |
| QPT | CVPR 2023 | 0.925 | 0.928 | - | - | 0.895 | 0.914 |
| ARNIQA | WACV 2024 | 0.908 | 0.912 | - | - | 0.880 | 0.901 |
| TOPIQ | TIP 2024 | 0.921 | 0.924 | 0.574 | 0.657 | 0.870 | 0.884 |
| CDINet | TMM 2024 | 0.920 | 0.919 | 0.865 | 0.880 | 0.898 | 0.908 |
| LoDa | CVPR 2024 | 0.876 | 0.899 | 0.932 | 0.944 | 0.869 | 0.901 |
| ADTRS | ICIP 2024 | - | - | 0.905 | 0.918 | 0.878 | 0.897 |
| VISGA | TCSVT 2025 | 0.919 | 0.925 | 0.930 | 0.937 | 0.901 | 0.914 |
| CoDI-IQA | ArXiv 2025 | 0.936 | 0.940 | 0.902 | 0.917 | 0.901 | 0.916 |
| DGIQA | ArXiv 2025 | 0.943 | 0.945 | **0.934** | **0.942** | 0.934 | 0.940 |
| RSFIQA | ArXiv 2025 | 0.953 | 0.954 | 0.934 | 0.940 | **0.951** | **0.959** |
| D2S-R18 | ours | 0.952 | 0.953 | 0.901 | 0.922 | 0.941 | 0.938 |
| D2S-R50 | | **0.958** | **0.959** | 0.900 | 0.925 | 0.938 | 0.935 |

performance, with D2S-R50 reaching 0.9541 SRCC and 0.9576 PCC, establishing new state-of-the-art results.

**Small Samples Training (SST).** We evaluate the few-shot behavior of D2S by randomly sampling 10, 50, 100, and 500 training images from IC9600 (Table 2). Performance rises steadily with sample size, particularly for SRCC and PCC. Notably, with only 10 samples, D2S-R50 reaches 0.6864 SRCC, significantly outperforming ICNet (0.5178), demonstrating that multi-modal alignment supplies strong semantic regularization even in extremely low-data regimes. With 50–100 samples, performance improves rapidly (e.g., 0.8680 at SST@100). With 500 samples, both D2S variants exceed 0.91 SRCC and PCC while reducing RMSE and RMAE. In contrast, ICNet and ICCORN degrade sharply with limited data. These results show that D2S inherits strong sample efficiency from its semantic teacher mechanism, making it attractive for engineering scenarios with scarce annotations.

**Cross-Dataset Generalization.** To evaluate generalization, we train D2S on IC9600 and directly test on Savoias, Nagle4k, and VISC-C/I without fine-tuning (Table 3). Across most metrics, both D2S-R18 and D2S-R50 surpass ICNet and ICCORN. On Nagle4k and VISC-C, D2S-R50 achieves SRCC values of 0.7976 and 0.7317, respectively, verifying that text-guided alignment enhances robustness under domain shifts. Although correlations on Savoias and VISC-C/I remain below 0.70, D2S still matches or slightly exceeds other supervised models.

Table 5: **Ablation study of the main components in D2S on IC9600.** Each case (a ∼ e) corresponds to different combinations of these modules.

| Case | AttnPool | EAL | FAL | SRCC ↑ | PCC ↑ | RMSE ↓ | RMAE ↓ |
|------|----------|-----|-----|--------|-------|--------|--------|
| (a) | × | × | × | 0.9396 | 0.9430 | 0.0547 | 0.2052 |
| (b) | ✓ | × | × | 0.9446 | 0.9476 | 0.0541 | 0.2050 |
| (c) | ✓ | ✓ | × | 0.9467 | 0.9503 | 0.0546 | 0.207 |
| (d) | ✓ | × | ✓ | 0.9473 | 0.9510 | 0.0551 | 0.2086 |
| (e) | ✓ | ✓ | ✓ | **0.9499** | **0.9540** | **0.0472** | **0.1906** |

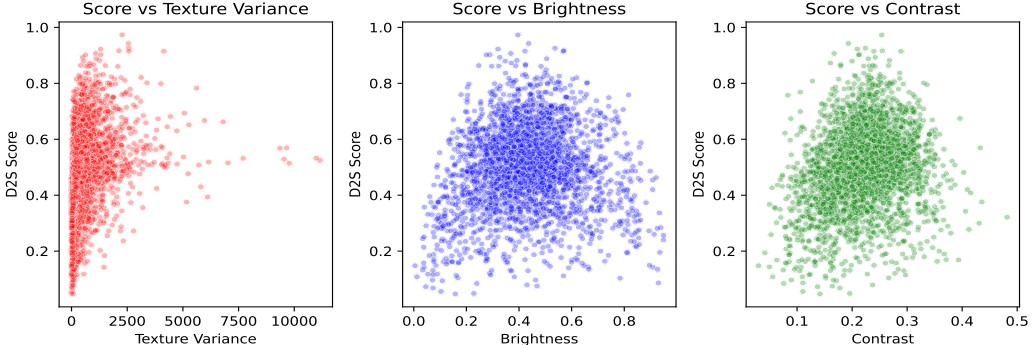

Figure 6: The correlation between the predicted scores of D2S and the low-level statistics, including texture variance, brightness, and contrast.

**Cross-Task Transfer.** We further assess transferability by applying D2S to NR-IQA (Table 4). Training and dataset details are provided in Appendix A.2 and A.3. D2S achieves competitive or superior performance compared with recent NR-IQA methods. On KADID-10K, D2S-R50 obtains 0.958 SRCC and 0.959 PLCC, the best among all compared methods. On KonIQ-10K, DGIQA remains the strongest, but D2S achieves competitive values near 0.90 SRCC. On TID2013, both variants exceed 0.938. These results show that semantic alignment not only benefits ICA but also transfers effectively to perceptual metrics that rely on visual–semantic coherence.

## 4.2 ANALYSIS OF D2S

**Ablation study of the main components.** We evaluate the effect of AttnPool, EAL, and FAL by incrementally adding each module (Table 5). AttnPool offers substantial improvement by enabling fine-grained feature aggregation. FAL provides moderate gains in correlation metrics, while EAL mainly improves error reduction and stability by controlling cross-modal entropy bias. When combined, D2S achieves the best performance (SRCC 0.9499, PCC 0.9540) with a significant drop in RMSE and RMAE. The complementary behavior of these modules highlights that semantic alignment must occur both at the representation level (FAL) and the distributional level (EAL) to maximize performance.

**What Does D2S Learn?** We analyze the top-20 frequently activated channels (Figure 7). D2S (top subplot) exhibits a concentrated and stable activation pattern, relying on a compact set of discriminative channels. These activations show no strong correlation with low-level image statistics such as brightness, contrast, or texture variance (Figure 6), indicating that D2S does not rely on superficial cues but instead captures higher-level patterns relevant to complexity.

Compared with a purely visual baseline (bottom subplot), D2S suppresses noisy or weakly informative channels and promotes more structured channel usage. This structured sparsity is a direct effect of introducing semantic supervision and also explains the improved cross-dataset generalization.

**Binned Error Analysis.** We divide predicted scores into 10 uniform complexity bins (Figure 8). D2S performs similarly to the visual-only baseline for simple images (complexity < 0.55), but con-

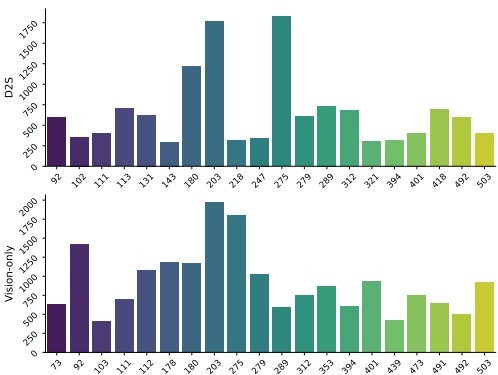

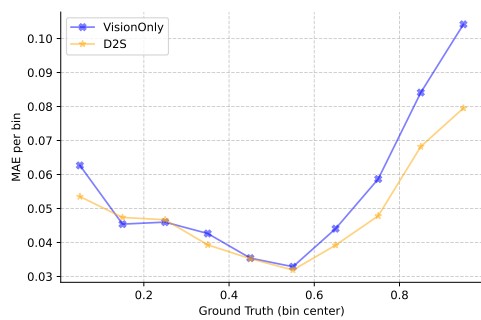

Figure 7: **Channel utilization histogram.** Frequencies of the top-20 most correlated feature channels across the IC9600 test set. The above subgraph represents D2S, and the one below is Vision-only.

Figure 8: **Binned MAE comparison.** Mean absolute error (MAE) across 10 uniformly spaced complexity bins.

sistently outperforms it as complexity increases. The margin widens for high-complexity scenes, confirming that textual guidance is most beneficial when structural and semantic cues become important.

**In addition, we provide the further analyses, ablations and discussions are in Appendix A.6 and A.9.** These include (1) sensitivity analyses on entropy temperature and buffer size, (2) robustness evaluation under caption perturbations, and (3) an extended study on semantic–visual alignment dynamics.

## 5 CONCLUSION

In this work, we addressed the challenge of image complexity assessment by introducing multimodal fusion into complexity modeling. We proposed the D2S framework, which leverages pretrained VLMs to describe images and integrates feature alignment and entropy alignment mechanisms to guide complexity assessment. Our theoretical analysis demonstrated that combining visual and semantic features enriches representation diversity and reduces the effective hypothesis space, thereby improving both accuracy and generalization. Extensive experiments validated these insights: D2S not only outperformed state-of-the-art methods on the IC9600 benchmark but also showed competitive transferability on image quality assessment datasets. Furthermore, the framework achieves this without incurring additional multi-modal inference cost, as only the visual branch is required at test time. These results show that semantic guidance offers a principled way to stabilize and regularize visual representations, yielding both performance gains and stronger cross-domain robustness. Taken together, these results highlight the effectiveness and efficiency of semantic alignment for complexity modeling and suggest broader applicability of text-guided representation shaping in perceptual understanding tasks beyond image complexity.

## 6 REPRODUCIBILITY STATEMENT

The detailed experimental settings of datasets, models, hyper-parameter settings, and computational resources can be found in Section 4 and Appendix A.2 and A.3. The codes, model checkpoints and `train_blip_caption.txt` for reproducing our main evaluation results are provided at `https://anonymous.4open.science/r/D2S-43A7`.

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

# A   APPENDIX

## A.1   THEORETICAL FOUNDATIONS AND EMPIRICAL ANALYSIS OF MULTIMODAL FUSION

### A.1.1   PROOF OF PROPOSITION 1

**Proof.** We first define the entropy of visual and semantic feature distributions. Assume the visual encoder outputs a feature distribution $\{p_k^{(v)}\}_{k=1}^{K}$, and the text encoder outputs a feature distribution $\{p_t^{(s)}\}_{t=1}^{T}$. Their corresponding entropies are defined as:

$$H_v^F(I) = -\sum_{k=1}^{K} p_k^{(v)} \log p_k^{(v)}, \qquad H_s^F(S) = -\sum_{t=1}^{T} p_t^{(s)} \log p_t^{(s)} \tag{14}$$

Here, $H_v^F(I)$ measures the diversity of the visual space (e.g., texture, color, structure), and $H_s^F(S)$ measures the diversity of the semantic space (e.g., object categories, relations, actions). To analyze multimodal fusion, we consider a linear entropy model:

$$H^F(I) = \alpha\, H_v^F(I) + \beta\, H_s^F(S) \tag{15}$$

where $\alpha, \beta > 0$ are weighting coefficients.

The expression above provides a tractable approximation of how visual and semantic entropies jointly contribute to fused representations. It does not imply that $H^F(I)$ must strictly exceed $H_v^F(I)$ for arbitrary choices of $\alpha$ and $\beta$. Instead, the goal is to capture how semantic cues can enrich the entropy of the fused representation under reasonable empirical settings.

In practice, for our trained model, we estimate $\alpha$ and $\beta$ by fitting the linear relation to the fused entropy measured from model activations. The fitted coefficients satisfy $\alpha > 0$ and $\beta > 0$, and the relation achieves high $R^2$ scores (e.g., $R^2 = 0.96$ for visual-only fusion and $R^2 = 0.58$ for vision–text fusion), indicating that $H^F(I)$ is well approximated by this linear form.

Empirical statistics also show that for the vast majority of samples (97.71%), the fused entropy $H^F(I)$ exceeds $H_v^F(I)$, with an average empirical slack of $K_{\text{emp}} \approx 0.2216$ and an estimated margin of 2.62. This suggests that semantic features typically contribute additional entropy to the fused representation, even though strict pointwise inequality is not guaranteed.

Since entropy is positively correlated with representational diversity and often correlates with perceived image complexity, these observations indicate that multimodal fusion tends to produce features that better capture the underlying complexity of the image. Let $C_v(I)$ denote the complexity inferred from the visual branch and $C_s(S)$ denote the contribution from the semantic branch. Under the linear model, we express fused complexity as:

$$C^F(I) = \alpha\, C_v(I) + \beta\, C_s(S) \tag{16}$$

While not enforcing universal monotonicity, the empirical increase in fused entropy suggests that $C^F(I)$ provides a more expressive approximation to the ground-truth complexity $C(I)$ than $C_v(I)$ alone. Thus, multimodal fusion offers a principled mechanism for enriching the representation space.

### A.1.2   EMPIRICAL ESTIMATION OF FUSION COEFFICIENTS

To validate the linear fusion model used in Proposition 1, we perform a regression analysis to estimate the coefficients $\alpha$ and $\beta$ directly from the fused entropy produced by our trained D2S model. Given $N$ training samples, we fit

$$H^F(I) \approx \alpha\, H_v^F(I) + \beta\, H_s^F(S) \tag{17}$$

using least squares. The resulting fits are:

$$\text{(1) Visual-only:} \quad \alpha_{\text{vis}} = 0.2486,\ \beta_{\text{vis}} \approx 0,\ R^2 = 0.96.$$
$$\text{(2) Vision–text fusion:} \quad \alpha_{\text{mul}} = 0.4490,\ \beta_{\text{mul}} = 1.1045,\ R^2 = 0.58.$$

The positive coefficients and reasonably high $R^2$ values demonstrate that the linear model captures the trend of entropy fusion in practice. Notably, $\beta_{\text{mul}} > 1$ suggests that semantic entropy contributes substantially to the fused entropy, consistent with the increased representational diversity observed in our experiments.

### A.1.3 Synthetic Validation of Entropy Fusion

To further validate Proposition 1 under controlled conditions, we conduct a synthetic experiment where the assumptions of the linear model hold exactly. We construct a high-rank Gaussian visual distribution with covariance $\Sigma_v$, a low-rank semantic distribution with covariance $\Sigma_s$, and a fused distribution $Z_f = \alpha Z_v + \beta Z_s$. Entropy for multivariate Gaussian is given by

$$H(Z) = \frac{1}{2} \log \left( (2\pi e)^d \det(\Sigma) \right) \tag{18}$$

By varying $\alpha$ and $\beta$ over a grid, we observe that $H(Z_f)$ increases monotonically with $\beta$, the increase is largest when $\Sigma_s$ contributes new orthogonal directions to $\Sigma_v$, the synthetic results match the empirical trends seen in real models.

This controlled experiment confirms that when semantic features add complementary variance directions, the fused entropy indeed grows, supporting the intuition behind Proposition 1.

### A.1.4 Text-induced Dimensional Reshaping

To support Theorem 2 (Section 3.3), we evaluate the intrinsic dimension (ID) of representations using the ID-PR estimator. We compute ID for baseline visual representation $Z_v$, text-only representation $Z_s$, and fused representation $\tilde{Z}_v$ after alignment.

We got $\text{ID}(Z_v) \approx 1.25$, $\text{ID}(Z_s) \approx 50$, $\text{ID}(\tilde{Z}_v) \approx 7.40$. These findings indicate the visual-only baseline collapses to a near one-dimensional manifold due to regression training. Semantic representations lie in a richer manifold. Fusion expands the visual representation to a moderate low-dimensional space rather than pure compression.

This clarifies the practical regime of D2S: semantic alignment reshapes the representation, improving expressivity while controlling dimensionality, consistent with the generalization benefits predicted by Rademacher complexity.

### A.1.5 Practical Notes and Limitations

We highlight two practical considerations:

**(1) Non-universality of monotonicity.** The inequality $H^F(I) > H_v^F(I)$ holds for most samples but is not mathematically universal. Caption noise, low-quality semantics, or very small $\beta$ may lead to exceptions. This aligns with our empirical analysis and does not weaken Proposition 1, which is framed as an approximate modeling principle.

**(2) Dependence on semantic quality.** If the text encoder produces low-entropy or irrelevant captions, the benefit to fused entropy diminishes. Our robustness experiments (Table 10) show that D2S remains stable under moderate perturbations but degrades with meaningless text inputs.

These considerations define the scope under which Proposition 1 and Theorem 2 are most informative.

### A.2 Implementation Details

We implement our model in PyTorch. The ResNet backbone is initialized with ImageNet-pretrained (Deng et al., 2009) weights from TIMM (Wightman, 2019), and the BLIP caption generator is employed in its large configuration. The CLIP text encoder is also initialized with pretrained parameters. We train the model using the Adam (Adam et al., 2014) optimizer with weight decay 1e-3. The initial learning rate was set to 1e-3 and the batch size to 32. The cosine annealing (Loshchilov & Hutter, 2016) was used for scheduling, with a minimum learning rate of 2.5e-6. The temperature was set to 0.07. Training was performed on a single NVIDIA RTX 3090.

For the results reported in Table 2, the training set was constructed by dividing the complexity score range $(0, 1)$ into ten intervals and randomly sampling within each interval. Specifically, the SST@10 setting samples one image from each interval, whereas SST@50 samples five images from each interval. It is worth noting that, due to the scarcity of high-complexity images, the number of intervals containing samples in SST@500 is fewer than fifty. Nevertheless, we refer to this configuration as SST@500 for consistency.

Table 6: **Full-data training results on Savoias, Nagle4K, and VISC-C.** Each dataset is split into training and testing sets with an 8:2 ratio, and all training hyperparameters follow those used in the main paper. **Bold** indicates the best overall performance. Underline indicates the best performance achieved by any single prompt. References: DBCNN (Network, 2022), NIMA (Talebi & Milanfar, 2018), CLIPIQA (Wang et al., 2023).

| Method | Savoias | | | Nagle4K | | | VISC-C | |
|---|---|---|---|---|---|---|---|---|
| | SRCC ↑ | PCC ↑ | | SRCC ↑ | PCC ↑ | | SRCC ↑ | PCC ↑ |
| DBCNN | 0.768 | 0.770 | | 0.745 | 0.732 | | 0.779 | 0.783 |
| NIMA | 0.781 | 0.771 | | 0.756 | 0.741 | | 0.810 | 0.803 |
| HyperIQA | 0.801 | 0.798 | | 0.772 | 0.758 | | 0.734 | 0.739 |
| CLIPIQA | 0.779 | 0.794 | | 0.785 | 0.774 | | 0.781 | 0.796 |
| TOPIQ | 0.838 | 0.832 | | 0.804 | 0.792 | | 0.803 | 0.811 |
| ICNet | 0.845 | 0.849 | | 0.815 | 0.804 | | 0.789 | 0.790 |
| ICCORN | 0.852 | 0.856 | | 0.818 | 0.806 | | 0.796 | 0.793 |
| D2S-R18 | 0.861 | 0.864 | | 0.823 | 0.813 | | 0.801 | 0.802 |
| D2S-R50 | **0.875** | **0.876** | | **0.834** | **0.825** | | **0.822** | **0.820** |
| | RMSE ↓ | RMAE ↓ | | RMSE ↓ | RMAE ↓ | | RMSE ↓ | RMAE ↓ |
| DBCNN | 0.147 | 0.355 | | 0.121 | 0.310 | | 0.086 | 0.351 |
| NIMA | 0.210 | 0.368 | | 0.134 | 0.315 | | 0.125 | 0.362 |
| HyperIQA | 0.293 | 0.392 | | 0.145 | 0.327 | | 0.181 | 0.388 |
| CLIPIQA | 0.171 | 0.348 | | 0.112 | 0.302 | | 0.122 | 0.348 |
| TOPIQ | 0.123 | 0.330 | | 0.101 | 0.290 | | 0.079 | 0.340 |
| ICNet | 0.121 | 0.325 | | 0.097 | 0.278 | | 0.123 | 0.349 |
| ICCORN | 0.119 | 0.324 | | **0.088** | 0.265 | | 0.119 | 0.342 |
| D2S-R18 | 0.128 | 0.321 | | 0.092 | 0.268 | | **0.115** | **0.328** |
| D2S-R50 | **0.114** | **0.301** | | 0.089 | **0.265** | | 0.116 | 0.332 |

For the results in Table 3, we adopted a cross-dataset evaluation protocol, where the full datasets from the ICA benchmark, including Nagle4k (Nagle & Lavie, 2020b), Savoias (Saraee et al., 2020b), and VISC-C/I (Kyle-Davidson et al., 2023), were used as test sets.

For the results in Table 4, we followed the common evaluation protocol in NR-IQA. After min-max normalizing the MOS scores of all images, each dataset was randomly partitioned into training, validation, and test subsets with a ratio of 6:2:2. We compared D2S with QPT (Zhao et al., 2023), ARNIQA (Agnolucci et al., 2024), TOPIQ (Chen et al., 2024), CDINet (Zheng et al., 2024), LoDa (Xu et al., 2024), ADTRS (Alsaafin et al., 2024), VISGA (Shi et al., 2025), CoDI-IQA (Liu et al., 2025b), DGIQA (Ramesh et al., 2025) and RSFIQA (Song et al., 2025).

## A.3 DATASETS

The datasets used in our study are drawn from two tasks: image complexity assessment (ICA) and image quality assessment (IQA). The ICA datasets include IC9600, Savoias, PASCAL VOC 4000 (Nagle4k), and VISC-C/I. Among them, IC9600 is employed to validate the effectiveness of our method, while Savoias, Nagle4k, and VISC-C/I are used to evaluate cross-dataset generalization. The IQA datasets, consisting of KADID-10K, KonIQ-10K, and TID2013, are utilized to assess the cross-task transferability of the proposed approach.

## A.4 EVALUATION METRICS

We adopt Spearman's Rank Correlation Coefficient (SRCC), Pearson's Linear Correlation Coefficient (PLCC), Root Mean Square Error (RMSE), and Relative Mean Absolute Error (RMAE) to evaluate the predictive performance of our model. In addition, we report the number of parameters (Params) and inference latency (Latency) to assess computational efficiency.

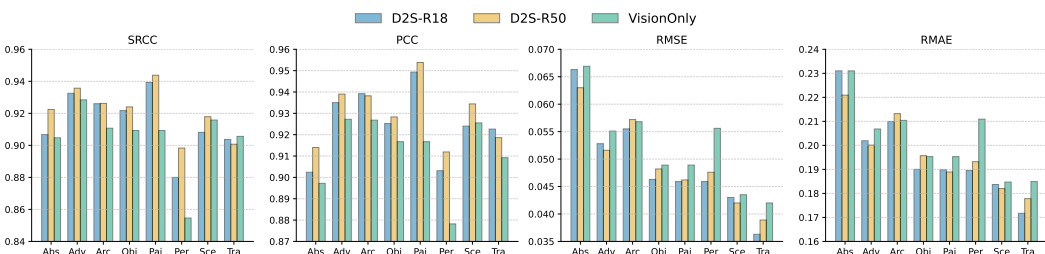

Figure 9: **Performance of semantic categories in IC9600.**

## A.5 EAL ALGORITHM

---

**Algorithm 1:** Entropy Distribution Alignment

---

**Input:** Image $I$, Caption $S$, D2S $f_\theta$, MoM $f_\xi$, Momentum $m$, FIFO Buffers $B_v, B_s$, Buffer size $M$, Refresh step $r$

**Before training:** Create $f_\theta$, copy as $f_\xi$; Create $B_v, B_s$ with $M$.

**for** *each mini-batch* **do**

    $f_\xi$ extract features $z_v, z_s$ via Eq.(10);

    Compute entropy $H_v, H_s$ via Eq.(14);

    Store into buffers $B_v$ and $B_s$, meanwhile remove oldest ones;

    **if** $B_v$ *and* $B_s$ *Entries number* $\geq \frac{M}{2}$ **then**

        Compute $\mathcal{L}_{\text{eal}}$ from $B_v$ and $B_s$;

    **else**

        $\mathcal{L}_{\text{eal}} = 0$

    Update $f_\theta$ via gradient;

    Update $f_\xi$ via $\xi \leftarrow m\xi + (1-m)\theta$;

    Refresh $r$ old entries via updated $f_\xi$;

---

## A.6 FURTHER ANALYSES AND ABLATIONS

### A.6.1 FULL-DATA TRAINING ON OTHER ICA BENCHMARKS.

We train D2S from scratch on Savoias, Nagle4K, and VISC-C. Table 6 shows that D2S-R50 consistently achieves the highest correlations and lowest errors on all three datasets, confirming that D2S generalizes well even without IC9600 pre-training.

### A.6.2 PERFORMANCE OF SEMANTIC CATEGORY IN IC9600.

We assess how semantic alignment affects different semantic categories in IC9600 by evaluating D2S-R18, D2S-R50, and the VisionOnly variant (Figure 9). D2S consistently surpasses VisionOnly across correlation (SRCC/PCC) and error metrics (RMSE/RMAE). The improvements are most pronounced in object-centric categories (*Obj*, *Per*) and abstract categories (*Abs*, *Pai*), where visual cues alone are insufficient. D2S-R50 further outperforms D2S-R18, especially in *Arc* and *Sce*, suggesting that larger visual backbones can better exploit semantic guidance. These results show that the proposed alignment strategy not only improves average accuracy but also stabilizes predictions across diverse semantic distributions.

### A.6.3 EAL AND FAL COEFFICIENT ABLATION

We fix ResNet18 as the backbone and vary the EAL coefficient while keeping FAL coefficient at 0.01, buffer size 2048, momentum 0.995, and refresh step 50. As shown in Figure 10, the best PCC (0.9548) occurs at EAL coefficient 5. Table 7 further shows the robustness of D2S to EAL hyperparameters. The performance variation remains small across a wide range of settings, indicating that EAL benefits primarily from relative rather than absolute entropy alignment strength.

Table 7: **Ablation study of EAL hyper-parameters on IC9600.** $M$: buffer size, tested with 512, 1024, 2048, 4096 entries. **mom.**: momentum for the MoM update, tested with 0.990, 0.995, 0.999, 0.9999. **steps**: buffer refresh step, tested with 16, 50, 128 iterations, corresponding to the number of entries updated per iteration (128, 40, 16, respectively). The training time is recorded on the A2000 GPU (h: hour).

| $M$ | mom. | steps | SRCC ↑ | PCC ↑ | RMSE ↓ | RMAE ↓ | Training time ↓ |
|---|---|---|---|---|---|---|---|
| 2048 | 0.99 | 50 | 0.9487 | 0.9533 | 0.0497 | 0.1955 | - |
| | 0.995 | | **0.9508** | **0.9545** | **0.0496** | **0.1962** | - |
| | 0.999 | | 0.9499 | 0.9540 | 0.0472 | 0.1906 | - |
| | 0.9999 | | 0.9484 | 0.9527 | 0.0497 | 0.1960 | - |
| 512 | 0.995 | 50 | 0.9494 | 0.9531 | 0.0519 | 0.2016 | **0.8h** |
| 1024 | | | 0.9493 | 0.9533 | 0.0516 | 0.2008 | 0.9h |
| 2048 | | | **0.9499** | **0.9540** | **0.0472** | **0.1906** | 1.1h |
| 4096 | | | 0.9501 | 0.9530 | 0.0529 | 0.2041 | 1.9h |
| 2048 | 0.995 | 16 | 0.9499 | 0.9539 | 0.0513 | 0.2008 | - |
| | | 50 | **0.9499** | **0.9540** | **0.0472** | **0.1906** | - |
| | | 128 | 0.9496 | 0.9539 | 0.0502 | 0.1973 | - |

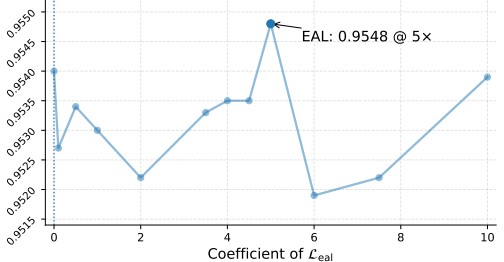
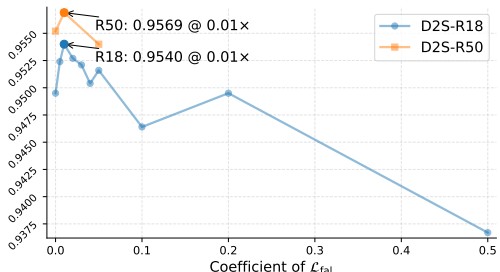

Figure 10: **EAL coefficient ablation.** D2S is trained for 20 epochs with ResNet18 as the backbone, while fixing the FAL coefficient at 0.01. The best result is achieved when the EAL coefficient equals 5.

Figure 11: **FAL coefficient ablation.** D2S is trained for 20 epochs with ResNet18 and ResNet50 as backbones, while fixing the EAL coefficient at 5. The best result is achieved when the FAL coefficient equals 0.01.

We vary the FAL coefficient while fixing EAL at its optimal configuration. Experiments using ResNet18 and ResNet50 show that the best performance is obtained at a FAL coefficient of 0.01 (Figure 11). Larger coefficients occasionally destabilize training due to excessive semantic pulling, while smaller values reduce the regularization effect.

### A.6.4 PROMPTS TEMPLATE AND CAPTIONS ABLATION

**Prompt Combination Matrix.** We test all single and multi-prompt combinations among the four prompts in Figure 5. Table 8 shows that no single prompt dominates. Instead, semantic complementarity is crucial. The full combination yields the best results.

**Impact of Different Caption Generators.** We further added results using Florence-2 and Qwen3-VL-4B captions for full D2S training (Table 9). Given the limited size of IC9600, stronger captioners do not significantly outperform BLIP. These results show that D2S is robust to caption noise and not tied to BLIP. Text functions as a soft semantic regularizer: richer semantics help, but the model does not collapse under weaker captions and does not require precise linguistic accuracy. This matches our theoretical view that alignment operates on distributional structure, so D2S only needs approximate semantics to reduce hypothesis-space ambiguity and guide the visual encoder.

**Effect of Incorrect or Corrupted Captions.** To quantify the robustness of D2S against degraded semantic supervision, we evaluate four forms of corrupted captions: full shuffling, partial shuffling,

Table 8: **Ablation of different prompt combinations.** All experiments are conducted on D2S-R18. Pi denotes the i-th prompt in Figure 5. [:, :] indicates the concatenation or combination of multiple prompts. **Bold** indicates the best overall performance. Underline indicates the best performance achieved by any single prompt.

| Prompts | SRCC ↑ | PCC ↑ | RMSE ↓ | RMAE ↓ |
|---|---|---|---|---|
| None | 0.9468 | 0.9495 | 0.0522 | 0.2010 |
| P1 | 0.9511 | 0.9533 | 0.0547 | 0.2082 |
| P2 | 0.9458 | 0.9505 | 0.0539 | 0.2053 |
| P3 | 0.9486 | 0.9520 | 0.0536 | 0.2053 |
| P4 | 0.9456 | 0.9507 | 0.0509 | 0.1986 |
| [P1, P2, P3] | 0.9487 | 0.9526 | 0.0521 | 0.2017 |
| [P1, P2, P4] | 0.9489 | 0.9527 | 0.0523 | 0.2019 |
| [P1, P3, P4] | 0.9485 | 0.9522 | 0.0517 | 0.2006 |
| [P2, P3, P4] | 0.9489 | 0.9532 | 0.0522 | 0.2018 |
| [P1, P2, P3, P4] | **0.9509** | **0.9544** | **0.0495** | **0.1962** |

Table 9: **Comparison of different caption generators.** These experiments are based on D2S-R18. BLIP and Qwen generate captions using the prompts shown in Figure 5.

| Captioner | Prompts | SRCC ↑ | PCC ↑ | RMSE ↓ | RMAE ↓ |
|---|---|---|---|---|---|
| Florence2 | caption | 0.9456 | 0.9501 | 0.0535 | 0.2044 |
| Florence2 | detailed caption | 0.9469 | 0.9506 | 0.0534 | 0.2044 |
| Florence2 | more detailed caption | 0.9441 | 0.9484 | 0.0524 | 0.2015 |
| BLIP Large | Figure 5 prompts | 0.9509 | 0.9544 | **0.0495** | **0.1962** |
| Qwen3-VL-4B | Figure 5 prompts | **0.9519** | **0.9552** | 0.0503 | 0.1976 |

random sentences, and fixed meaningless words. Table 10 shows that mild corruption produces only small drops, while removing semantics entirely (case d) leads to a clear decline, confirming that semantic structure (rather than mere text tokens) drives the improvement in D2S.

**Performance Using Only Captions.** To disentangle the contribution of textual information, we train models using captions alone without image input. As shown in Table 11 (left), performance ranges from 0.7095 (CapI) to 0.8260 (BLIP) in SRCC. This confirms that captions contain meaningful complexity cues but cannot fully capture visual diversity, reinforcing the complementary nature of multimodal fusion.

### A.6.5    PROJECTION AND ALIGNMENT ABLATION

**Alternative Alignment Strategies.** We further evaluate alternative forms of alignment, including CLIP-style captions, a CLIP-pretrained aligner, and BLIP textual embeddings without prompt generation. As shown in Table 12, most alternatives remain competitive, while BLIP textual embeddings slightly outperform the default alignment. This confirms that the alignment mechanism is flexible and not tied to a specific captioner.

**Projection Layer Ablation.** We test whether the linear projection layer in the visual branch is necessary. Removing it results in slight increases in RMSE and RMAE (Table 13), indicating that even a simple linear mapping helps stabilize cross-modal alignment. However, this impact is so minor that it can be safely ignored. It is merely the alignment of the same latitude that enables D2S to converge more quickly (D2S surpassed ICNet in just 20 epochs).

**Visualization of modality alignment.** We visualize visual and textual embeddings using t-SNE before and after alignment. In Figure 12, the two modalities form separate clusters. After alignment (Figure 13), the clusters become interleaved, indicating that the alignment module effectively bridges the semantic gap. This qualitative behavior matches the entropy statistics in Figure 14, where the overlap between visual and textual entropy distributions increases substantially after EAL.

Table 10: **Effect of incorrect captions on IC9600 performance.** '−' denotes the original set of four prompts used in Figure 5. 'm1.0' indicates that all words are fully shuffled, and 'm0.5' indicates that half of the words are shuffled. 'rs' refers to grammatically valid random sentences. 'wm' refers to meaningless token sequences such as 'any any any ...'

| Case | Method | SRCC ↑ | PCC ↑ | RMSE ↓ | RMAE ↓ |
|------|--------|--------|-------|--------|--------|
| – | | **0.9509** | **0.9544** | **0.0495** | **0.1962** |
| a | m1.0 | 0.9479 | 0.9518 | 0.0517 | 0.2011 |
| b | m0.5 | 0.9508 | 0.9539 | 0.0512 | 0.1996 |
| c | rs | 0.9396 | 0.9409 | 0.0559 | 0.2076 |
| d | mw | 0.9238 | 0.9242 | 0.0851 | 0.2645 |

Table 11: **Comparisons with different captions.** 'Only Caption' indicates using captions alone for IC evaluation. 'Image & Caption Concat' means concatenating the image features and text features before inputting them into the regression head. SRCC and PCC, larger ↑ is better. RMSE and RMAE, smaller ↓ is better.

| CapType | Only Caption | | | | Image & Caption Concat | | | |
|---------|------|------|------|------|------|------|------|------|
| | SRCC | PCC | RMSE | RMAE | SRCC | PCC | RMSE | RMAE |
| CapI | 0.7095 | 0.7162 | 0.1084 | 0.2868 | 0.9456 | 0.9501 | 0.0535 | 0.2044 |
| CapII | 0.8120 | 0.8172 | 0.0903 | 0.2630 | 0.9469 | 0.9506 | 0.0534 | 0.2044 |
| CapIII | 0.8102 | 0.8151 | 0.0904 | 0.2626 | 0.9441 | 0.9484 | 0.0524 | **0.2015** |
| BLIP | **0.8260** | **0.8251** | **0.0888** | **0.2609** | **0.9476** | **0.9524** | **0.0524** | 0.2029 |

**Prediction scatter and regression line.** Scatter plots in Figure 15 show that D2S predictions lie closer to the ideal regression line than the baseline. This indicates not only stronger correlation but also reduced systematic bias. The tighter dispersion reflects the stabilizing effect of semantic regularization, which improves local ranking consistency across complexity levels.

**Feature Activation Visualization.** We further analyze how D2S attends to image regions by comparing maximum activation, mean activation, and Grad-CAM visualizations (Figure 16). Mean activation maps highlight the largest number of structural regions, offering a more global representation of complexity. Maximum activation maps capture salient object parts, while Grad-CAM produces sparser responses and tends to miss fine-grained details. Under semantic guidance, D2S exhibits concentrated activation near caption-related objects, indicating that textual cues steer the visual encoder toward semantically meaningful structures.

## A.7 RELATED WORKS

### A.7.1 IMAGE COMPLEXITY ASSESSMENT.

**Statistical features.** Early methods for image complexity assessment primarily relied on statistical features or low-level visual indicators, such as entropy (Chikhman et al., 2012), symmetry (Chipman, 1977; Kyle-Davidson et al., 2022), spatial layout (Olivia et al., 2004), and compressibility (Palumbo et al., 2014; Machado et al., 2015). These approaches are easy to implement and highly interpretable, but they suffer from clear limitations. Specifically, they mainly capture local structures and texture details, while being sensitive to noise and image resolution.

**Deep learning models.** With the advent of deep learning, several approaches attempted to directly model image complexity using learning-based frameworks (Nagle & Lavie, 2020a; Saraee et al., 2020a; Kyle-Davidson et al., 2022). For instance, ICNet (Feng et al., 2022) improves complexity regression by combining multi-scale inputs with convolutional features, while ICCORN (Guo et al., 2023) employs a larger backbone and integrates ordinal regression constraints to enhance perceptual modeling. Celona et al. (Celona et al., 2024) further introduced ViTs for complexity assessment, highlighting the potential of deep features in complexity modeling. Nevertheless, these models largely focus on low-level image cues related to complexity, overlooking the human tendency to rely on high-level semantic information when making judgments.

Table 12: **Evaluation of alternative alignment methods.** '−' denotes the D2S-R18 model reported in the main paper. 'CLIP' denotes captions generated by BLIP in a CLIP-style format such as 'a photo of . . . '. 'aligner' denotes a two-branch alignment module, where each branch contains three linear layers and is attached to the CLIP-based aligner trained on IC9600. 'embed' denotes skipping caption generation and directly aligning BLIP textual embeddings with D2S visual features during training.

| Case | Method | SRCC ↑ | PCC ↑ | RMSE ↓ | RMAE ↓ |
|---|---|---|---|---|---|
| – | | 0.9509 | 0.9544 | **0.0495** | **0.1962** |
| a | CLIP | 0.9452 | 0.9493 | 0.0563 | 0.2107 |
| b | aligner | 0.9464 | 0.9502 | 0.0506 | 0.1967 |
| c | embed | **0.9510** | **0.9545** | 0.0504 | 0.1977 |

Table 13: **Ablation of the projection layer in D2S-R18.** This experiment examines whether the lightweight linear projection layer, which maps visual features into the joint vision–text latent space, is necessary for effective semantic alignment. Removing the projection layer forces the visual features to interact with textual features in their original spaces, thereby weakening the geometric consistency between modalities.

| Method | SRCC ↑ | PCC ↑ | RMSE ↓ | RMAE ↓ |
|---|---|---|---|---|
| D2S-R18 w/ projection | 0.9509 | 0.9544 | **0.0495** | **0.1962** |
| D2S-R18 w/o projection | **0.9506** | **0.9545** | 0.0519 | 0.2016 |

**High-level semantics.** To our knowledge, Shen et al. (Shen et al., 2024) were the first to introduce high-level semantics into this field. They quantified the number of segments and categories in an image using SAM (Kirillov et al., 2023) and FC-CLIP (Yu et al., 2023), respectively, and employed a linear regression model to predict complexity scores. This improved interpretability, but yielded suboptimal performance. Li et al. (Li et al., 2025) argued that implicit motion in image objects could benefit ICA, leveraging VLMs to generate simple captions that served as prompts to convert static images into dynamic videos. By fusing video, image, and text branches, they achieved the best performance to date on IC9600, but at the cost of ∼11B parameters and substantial computational resources.

**Our approach.** In contrast, our approach extracts high-level semantic information from VLMs through carefully designed prompt templates, using it only to guide the image branch during training. **At inference, our model requires only the visual input.**

A.7.2 VISION-LANGUAGE MODELING.

In recent years, visual-language models (VLMs) have achieved remarkable progress in tasks such as image captioning (Li et al., 2022), cross-modal retrieval (Wang et al., 2025), and visual question answering (Kuang et al., 2025), with representative models including CLIP, BLIP, and ALIGN (Jia et al., 2021). By aligning vision and language representations, these models effectively capture the semantic information of objects, relationships, and scenes. Prior studies show that language descriptions provide complementary information beyond low-level features, enhancing a model's ability to understand and quantify image content. However, research on visual-language alignment mechanisms for image complexity modeling remains limited. Current complexity assessment methods have not fully exploited semantic information to improve cross-domain generalization. This gap motivates our Describe-to-Score framework, which achieves unified modeling of low-level vision and high-level semantics through visual-language fusion, thereby enhancing both the accuracy and generalization of complexity assessment.

A.8 IMAGE CAPTION EXAMPLES

In Figure 17 and 18, we present some examples of captions. We selected one image-caption pair from each of the ten score ranges. The prompt template text is marked in green, while the generated text is marked in black or red. We found that the last generated text was more incorrect (in red). We

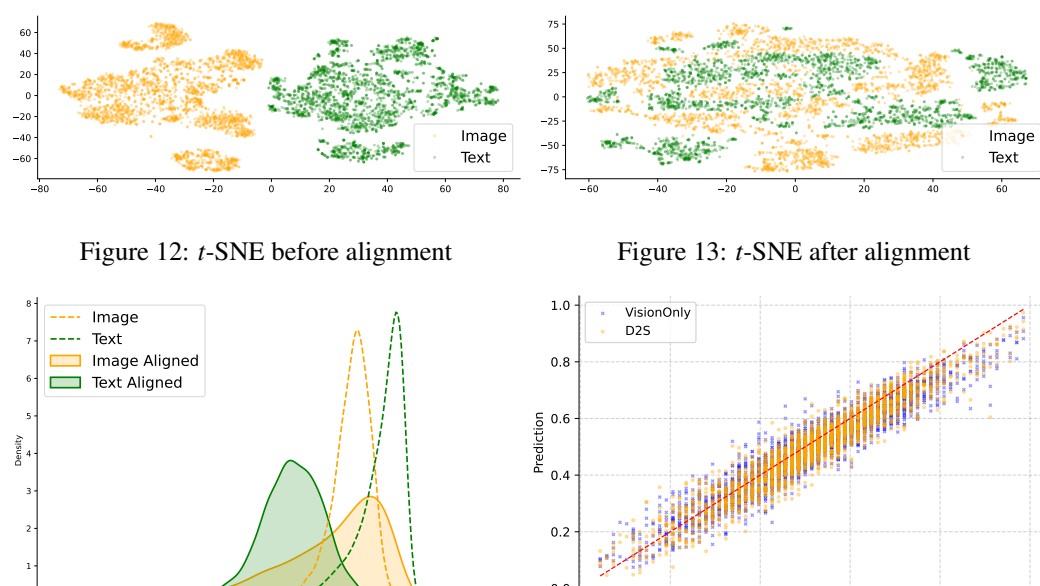

Figure 12: *t*-SNE before alignment

Figure 13: *t*-SNE after alignment

Figure 14: **Entropy distribution alignment.** Entropy distributions of visual and textual modalities before (dashed lines) and after alignment (solid lines).

Figure 15: **Prediction scatter plot.** Scatter plots of predicted scores versus ground-truth labels.

believe this might be due to the overly abstract language. VLMs are unable to generate features in the image that match them, resulting in unexpected answer.

## A.9 DISCUSSION

**Does ICA really require high-level semantic information?**

A fundamental question in this work is whether high-level semantic information is necessary for accurate image complexity assessment. Low-level visual cues such as texture, color, and edge density provide a baseline measure of complexity, but human perception may also rely on semantic content, which can potentially improve prediction accuracy.

Table 14 compares the performance of multiple methods across three modalities: Text-only (CapI, BLIP), Vision-only (CLICv2, HyperIQA, ICNet, ICCORN), and Vision-Text fusion (MICM, D2S R18/R50). Text-only methods rely purely on image captions. CapI achieves PCC 0.716 and BLIP improves to 0.825, indicating that textual cues capture part of the complexity information but are insufficient for high-precision prediction. Vision-only methods leverage visual features, with unsupervised methods like CLICv2 (PCC 0.870) learning content-invariant complexity representations. Supervised visual models reach PCC 0.949, highlighting that low-level visual features dominate complexity perception and provide strong baseline performance. Vision-Text fusion methods integrate both visual and semantic cues. MICM achieves PCC 0.953, and D2S-R50 reaches 0.958, with corresponding RMSE and RMAE showing the lowest prediction errors. Compared to vision-only baselines, this demonstrates a clear improvement, suggesting that semantic guidance provides measurable benefits in complexity assessment.

**Observations from these results.** Textual information alone is limited. While caption-based models partially capture complexity cues (PCC 0.716–0.825), they cannot match vision-based performance. Vision dominates baseline performance. Supervised vision-only models already achieve high PCC (0.947–0.949), confirming that low-level visual features are sufficient for most complexity signals. Semantic cues refine predictions. Vision-text fusion models consistently outperform vision-only methods, albeit the improvement is smaller than the jump from text-only to vision-only.

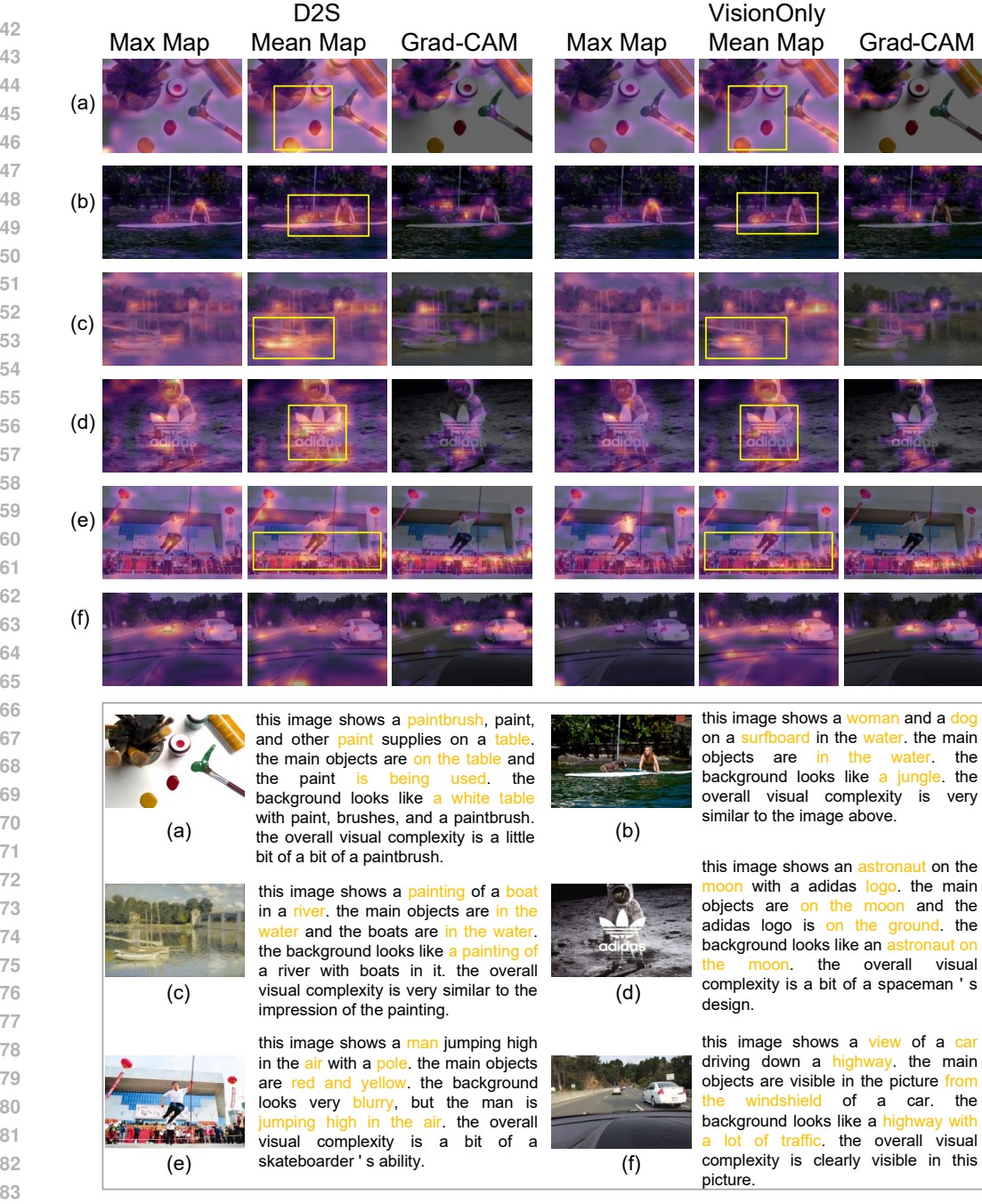

Figure 16: **Feature activation maps.** Visualization of channel activation maps (stage 4 in ResNet) using three approaches: maximum channel activation, mean channel activation, and Grad-CAM.

This indicates that high-level semantics are not strictly necessary but provide a subtle enhancement, particularly for fine-grained correlation with human perception.

Therefore, the progression from Text-only to Vision-only to Vision-Text fusion highlights that while low-level visual features are the primary driver of complexity prediction, high-level semantic information can further refine accuracy. D2S exemplifies this synergy, leveraging semantic cues to enhance performance without replacing the foundational value of visual representation.

Table 14: Comparison of complexity predictions for different modalities. All the results are derived from the aforementioned table (rounded to three decimal places).

| Method | Modal | SRCC ↑ | PCC ↑ | RMSE ↓ | RMAE ↓ |
|---|---|---|---|---|---|
| CapI | Text-only | 0.710 | 0.716 | 0.108 | 0.287 |
| BLIP | | 0.826 | 0.825 | 0.089 | 0.261 |
| CLICv2 | Vision-only | 0.879 | 0.870 | - | - |
| HyperIQA | | 0.935 | 0.935 | 0.067 | 0.229 |
| ICNet | | 0.945 | 0.947 | 0.058 | 0.216 |
| ICCORN | | 0.946 | 0.949 | 0.053 | 0.209 |
| MICM | Vision-Text fusion | 0.943 | 0.953 | 0.060 | - |
| D2S-R18 | | 0.951 | 0.954 | 0.050 | 0.196 |
| D2S-R50 | | **0.954** | **0.958** | **0.050** | **0.196** |

## A.10 THE USE OF LARGE LANGUAGE MODELS (LLMS)

We used ChatGPT-5 solely for polishing and grammar-checking the English prose. No content, ideas, or experimental decisions were generated or influenced by any LLMs.

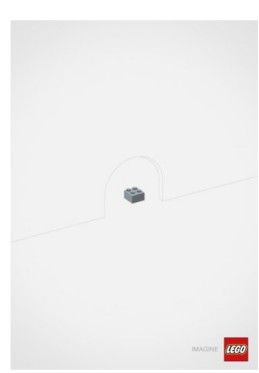

**Score: 0.05882352963089943**

*this image shows* a lego brick on a white background. *the main objects are* arranged in a single line on the white background. *the background looks* like a lego building with a small window. *the overall visual complexity is* a bit of a lego.

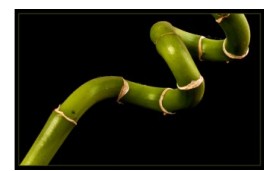

**Score: 0.1617647111415863**

*this image shows* a close up of a green plant with a black background. *the main objects are* green and the stems are brown. *the background looks* like a black background with a picture of a green plant. *the overall visual complexity is* very similar to the bamboo plant.

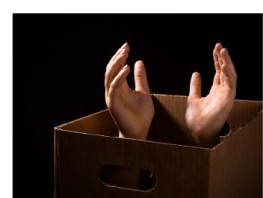

**Score: 0.25**

*this image shows* a person ' s hands in a box with a hole in it. *the main objects are* in a box with two hands. *the background looks* black and dark, but the hands are visible. *the overall visual complexity is* a very important feature in this photograph.

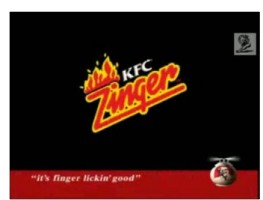

**Score: 0.3529411852359772**

*this image shows* a picture of a baseball game with the name of the team. *the main objects are* in the game and the text is in the background. *the background looks* like a fire with a baseball and a ball. *the overall visual complexity is* clearly visible in the title screen of the game.

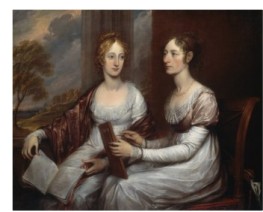

**Score: 0.45588234066963196**

*this image shows* a painting of two women sitting on a chair. *the main objects are* in the painting. *the background looks* like a painting of two women sitting in a chair. *the overall visual complexity is* very similar to the painting above.

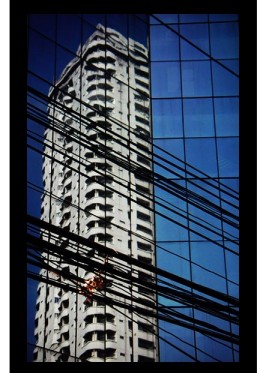

**Score: 0.5588235259056091**

*this image shows* a tall building with many windows and wires. *the main objects are* reflected in the windows of the building. *the background looks* like a reflection of a building in a mirror. *the overall visual complexity is* clearly visible in this picture.

Figure 17: Image-caption pair examples (part 1).

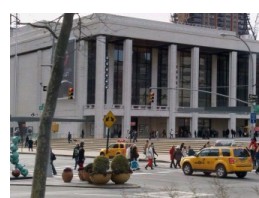

**Score: 0.6617646813392639**

*this image shows* a busy intersection with a taxi and pedestrians. *the main objects are* in the foreground of the busy intersection. *the background looks* like a city with a lot of people walking around. *the overall visual complexity is* a bit of a yellow cab.

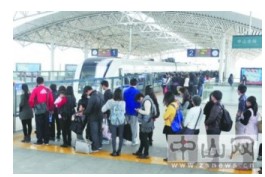

**Score: 0.75**

*this image shows* a group of people waiting at a train station. *the main objects are* in the station and people are waiting. *the background looks* like a train station with people waiting for the train. *the overall visual complexity is* very important to the image.

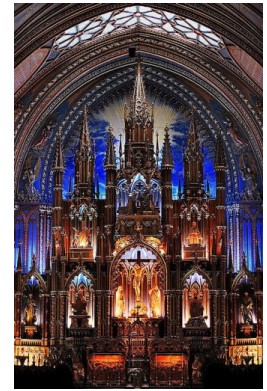

**Score: 0.8529411554336548**

*this image shows* a church with a stained glass ceiling and a clock. *the main objects are* lit up in a church with a stained glass ceiling. *the background looks* like a painting of a church with a stained glass window. *the overall visual complexity* is very similar to the actual church.

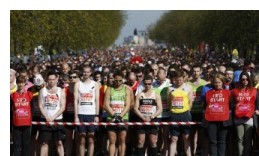

**Score: 0.9411764740943909**

*this image shows* a large group of people standing in a line. *the main objects are* lined up in a line at the start of a race. *the background looks like* a crowd of people waiting for the start of a marathon. *the overall visual complexity is* a key factor for the marathon.

Figure 18: Image-caption pair examples (part 2).

