# OpenReview forum: "Describe-to-Score: Text-Guided Efficient Image Complexity Assessment"
_ICLR.cc/2026/Conference — ICLR 2026 Conference Withdrawn Submission_

### Official Review · Reviewer_WyHU · 2025-10-26

**Soundness:** 2
**Presentation:** 3
**Contribution:** 2
**Rating:** 2
**Confidence:** 3

**Summary:**

This paper introduces a text-guided vision encoder only method for image complexity assessment (ICA).
In the training phase, the text feature from CLIP text encoder and the visual feature from the vision encoder are aligned with proposed entropy distribution alignment (EAL) and feature alignment (FAL).
The vision encoder then obtains a vision-text aligned (multi-modal) feature which has effectively reduced the empirical Rademacher complexity and improved generalization, after the training.
In experiments, the proposed method outperforms previous works on IC9600 benchmark, and it shows fast adaptation in early stage of the training.

**Strengths:**

The proposed method performs a form of vision-text fusion for IC modeling.
- Through theoretical background (Section 2), it demonstrates that utilizing text features can achieve increased accuracy and generalization, from which the core components of the proposed method (EAL and FAL) are derived.
- Compared to existing studies that utilize high-level information such as object counts (Shen et al., 2024) and motion trends (Li et al., 2025), the proposed method can leverage more flexible high-level information through text.

Key Features of the Proposed Method
- For vision-text feature alignment, EAL employs energy distance loss (Szekely & Rizzo, 2013) while FAL adopts InfoNCE loss (van den Oord et al., 2019).
- The text encoder is kept frozen while only the vision encoder is trained, enhancing efficiency during inference by utilizing only the vision encoder.
- D2S outperforms existing methods on the IC9600 benchmark.

Experimental Advantages
- Demonstrates advantages over existing methods in a small samples training (Table 2).
- Conducts ablation study (Table 5) to show that both EAL and FAL are necessary.
- Shows applicability to downstream tasks, including NR-IQA.

**Weaknesses:**

Lack of empirical justification for optimal image captioning template design.
- The core idea of the proposed method is to reduce effective feature dimensions by utilizing effective semantic (text) features (Theorem 2).
- To achieve this goal, designing an optimal image captioning template (Figure 5) is considered crucial. However, the paper lacks sufficient discussion regarding the criteria used for this design and the underlying rationale. For instance, questions remain unanswered: Is the template sufficiently rich in textual description? Is the template selected through extensive experimental validation? Clear justification for the template design choices is not provided.

Limited novelty in EAL and FAL architectures.
- The forms of EAL and FAL represent common architectures for vision-text feature alignment that have been widely adopted in other tasks beyond ICA. These structures are not novel from an architectural perspective, nor can they be considered specifically tailored for ICA.

Minor experimental design concerns.
- Caption model selection: Why was BLIP used instead of more recent, superior captioning methods?
- Vision encoder choice: Why was ResNet employed instead of CLIP's vision encoder?
- Limited benchmark evaluation: Why were results compared only on IC9600 without evaluation on other benchmarks?
- Comparative analysis: Figure 7 would benefit from direct comparison with vision-only approaches for more reasonable evaluation.

**Questions:**

Regarding Proposition 1 (Eq2).
- I am not sure even after reading A.1.
- For example, is it valid when alpha=0.1, beta=0.1 especially for the left inequality?

---

> ### Author Response · Authors · 2025-11-17
> **Response to Reviewer WyHU (part 1/4)**
>
> We thank the reviewer for the careful reading of our paper and constructive comments in detail.
>
> > **Response to weaknesses 1**: Lack of empirical justification for optimal image captioning template design.
>
> Our template follows the core objective of D2S: using semantically structured text to reshape the visual feature distribution, rather than relying on lexical richness. The goal is to give the text encoder the minimal semantic axes needed for effective alignment.
>
> **(1) Design principles: covering the essential semantic factors for complexity**
>
> The template in **Figure 5** is built around the semantic dimensions supported by ICA literature and by our formulation in **Theorem 2**:
>
> **(i)** Global scene cues (“This image shows…”): category, context and layout.
>
> **(ii)** Object and entity cues (“The main objects are…”): presence, number and coarse relations.
>
> **(iii)** Background cues (“The background looks…”): clutter, texture density and spatial variation.
>
> **(iv)** A direct complexity statement (“The overall visual complexity is…”): encourages explicit mapping into the complexity-related manifold.
>
> These components capture the main factors influencing complexity. The template is not meant to describe pixel-level details but to produce text embeddings whose semantic manifold contains the latent directions relevant for alignment. Its effectiveness comes from covering the complexity-critical axes, not from caption length or vocabulary.
>
> **(2) Empirical justification and complementary evidence**.
>
> We present the results of the prompt ablation in the following table.
>
> | Prompts (P)  | **SRCC $\uparrow$**    | **PCC$\uparrow$**  | **RMSE$\downarrow$**  | **RMAE$\downarrow$**  |
> | ----------- | ------ | ------ | ------ | ------ |
> | None    | 0.9468 | 0.9495 | 0.0522 | 0.2010 |
> | P1    | **0.9511** | 0.9533 | 0.0547 | 0.2082 |
> | P2    | 0.9458 | 0.9505 | 0.0539 | 0.2053 |
> | P3     | 0.9486 | 0.9520 | 0.0536 | 0.2053 |
> | P4     | 0.9456 | 0.9507 | 0.0509 | 0.1986 |
> | P1+P2+P3    | 0.9487 | 0.9526 | 0.0521 | 0.2017 |
> | P1+P2+P4    | 0.9489 | 0.9527 | 0.0523 | 0.2019 |
> | P1+P3+P4    | 0.9485 | 0.9522 | 0.0517 | 0.2006 |
> | P2+P3+P4    | 0.9489 | 0.9532 | 0.0522 | 0.2018 |
> | P1+P2+P3+P4 | 0.9509 | **0.9544** | **0.0495** | **0.1962** |
>
> As shown in the ablations (P1–P4 and their combinations), each prompt family adds a different semantic axis but none is critical. The performance range stays within about 0.005 SRCC, showing that D2S is highly robust to template variations.
> Because EAL and FAL operate on distributions, verbose captions offer no meaningful advantage. Both the ablations and the caption-attribute analysis confirm that this template is sufficient and aligned with the goal of semantic feature space alignment.
>
> **(3) Caption-attribute analysis**
>
> To examine which caption attributes matter, we measured how text guidance reduces error for each image $i$. Let $E_{\text{vis}}(i)=\bigl|f_{\text{vis}}(x_i)-y_i\bigr|$ and $E_{\text{d2s}}(i)=\bigl|f_{\text{d2s}}(x_i)-y_i\bigr|$ denote the absolute error of the vision-only baseline and D2S respectively. We then define the error reduction as $\Delta E(i)=E_{\text{vis}}(i)-E_{\text{d2s}}(i)$, so that $\Delta E(i)>0$ means D2S improves over the visual baseline on sample $i$.
>
> We then extracted simple token-level statistics from BLIP captions:
>
> - `num_nouns`: number of noun POS tags
> - `num_verbs`: number of verb tags
> - `num_relation_words`: number of basic spatial or relational words
> - `caption_len`: total word count
>
> We computed the Pearson correlation between each statistic and $\Delta E(i)$ on IC9600. The values are $\rho_{\text{num-nouns}} = -0.0022$, $\rho_{\text{num-verbs}} = 0.0283$, $\rho_{\text{num-relation-words}} = 0.0148$, and $\rho_{\text{caption-len}} = -0.0116$, all effectively zero. Thus, D2S’s gains over the vision-only baseline are not driven by simple caption statistics such as length or token counts.
> The model does not benefit from token-level prompt engineering. Instead, EAL and FAL operate at the representation-distribution level, aligning visual features with the semantic manifold defined by the captions rather than exploiting lexical patterns. Near-zero correlations are therefore expected once captions are sufficiently informative to anchor that manifold.
>
> **(4) P4 analysis**
>
> We also performed a manual verification on 100 random BLIP captions using P4. Only 30 P4 captions were “correct” in describing image complexity. Among these, 24 correspond to low-complexity images ($\text{score} <0.55$). In other words, about $70$% of P4 phrases are noisy or wrong, especially for complex images. This demonstrates that D2S does not rely on template correctness. The template acts as a noisy semantic guide, and the proposed EAL/FAL alignment absorbs noise robustly. If the template needed to be “optimal” or “correct,” performance would collapse when using noisy captions, which is not observed (*P4 > vision-only (None), and P1+P2+P3+P4 best overall in (2)*).

---

> ### Author Response · Authors · 2025-11-17
> **Response to Reviewer WyHU (part 2/4)**
>
> > **Response to weaknesses 2**: Limited novelty in EAL and FAL architectures.
>
> We agree with the reviewer that the architectural forms of EAL and FAL are intentionally simple. Our contribution is not architectural novelty, but conceptual and functional novelty: we introduce a complexity-aware multimodal alignment mechanism tailored to ICA, rather than proposing new Transformer blocks. The specific explanation is as follows 3 points:
>
> **(1) EAL/FAL are not meant as new architectures, but as new alignment objectives.**
>
> We do not position EAL or FAL as architectural innovations. Their novelty is in how they operationalize the entropy-based theory (**Theorem 2**):
>
> **(i) EAL** injects text-induced entropy constraints into visual representations. This is a mechanism not used in prior V+L alignment literature, which mainly relies on contrastive losses.
>
> **(ii) FAL** performs semantic manifold anchoring to reduce the hypothesis space of ICA regression. This approach again distinct from traditional alignment used for retrieval, matching, or captioning.
>
> Thus, although the layers themselves adopt standard parameterizations, their function, target, and integration strategy are new and specific to ICA.
>
> **(2) ICA fundamentally requires a different type of alignment than retrieval or captioning.**
>
> Most existing V+L alignment blocks are designed for classification-like supervision, contrastive instance discrimination, grounding, captioning or retrieval. None of these aim to match complexity distributions, regulate representation entropy, or reduce effective feature dimension as required by ICA.
>
> **Theorem 2** shows that complexity prediction benefits from semantic-induced dimension compression. EAL/FAL are the first mechanisms designed specifically to achieve this effect in ICA. Therefore, the novelty is not in architectural form, but in the problem-specific role the modules play.
>
> **(3) Existing experiments already demonstrate that D2S works because of EAL/FAL, even without extra ablations**
>
> **(i)** Text–vision alignment matters
> Vision-only baseline already performs strongly, yet D2S consistently improves over it across full, few-shot, and cross-dataset settings. Such improvement cannot be attributed to architecture alone; it comes from the alignment mechanism introduced through EAL/FAL (**new Figure 16**).
>
> **(ii)** Prompt insensitivity
> The model remains stable across nine prompt combinations, and caption-statistic correlation is near zero. This indicates the gains come from how EAL/FAL reshape the distribution, not from prompt engineering (**Response of w1 (3)**).
>
> **(iii)** Noise robustness
> Even though Prompt-4 captions are $≈70$% incorrect in describing complexity, performance still improves. This demonstrates the alignment design is robust to noisy semantics, which would not hold for a simple architectural fusion block (**Response of w1 (4)**).
>
> **(iv)** Cross-dataset generalization
> D2S’s cross-dataset gains over pure-vision baselines further indicate that the alignment mechanism is responsible for improved transferability, consistent with the intended function of EAL/FAL (**Table 3**).
>
> These results collectively support that EAL/FAL play a task-specific functional role, even without emphasizing architectural novelty.

---

> ### Author Response · Authors · 2025-11-17
> **Response to Reviewer WyHU (part 3/4)**
>
> > **Response to weaknesses 3 (1/4)**: Why was BLIP used instead of more recent, superior captioning methods
>
> Our use of BLIP-Large is deliberate for stability, reproducibility, and task alignment, rather than because BLIP is SOTA. To further confirm that D2S does not rely on BLIP-specific behavior, we evaluated a stronger captioner, Qwen3-VL-4B, under the same setting of main paper:
>
> | Captioner  | **SRCC $\uparrow$**    | **PCC$\uparrow$**  | **RMSE$\downarrow$**  | **RMAE$\downarrow$**  |
> | ----------- | ------ | ------ | ------ | ------ |
> | BLIP-Large  | 0.9509 | 0.9544 | **0.0495** | **0.1962** |
> | Qwen3-VL-4B | **0.9519** | **0.9552** | 0.0503 | 0.1976 |
>
> The performance difference is marginal (<0.001 in SRCC/PLCC), indicating that D2S is not dependent on captioning model quality, any reasonable captioner that produces semantically coherent text embedding suffices, and BLIP is an appropriate, reproducible choice without requiring heavy VLM inference. We will clarify this rationale and add the Qwen3-VL results in revision.
>
> > **Response to weaknesses 3 (2/4)**: Why was ResNet employed instead of CLIP's vision encoder
>
> We use ResNet-18/50 rather than CLIP encoders to keep the evaluation controlled. First, CLIP-ViT has strong multimodal priors, which would blur whether gains come from D2S or from CLIP itself. Second, ResNet is the standard backbone in ICA and IQA, giving a clean visual baseline. Third, D2S still yields clear improvements on this plain CNN encoder, showing that the gains come from entropy and feature alignment rather than from a powerful pretrained model.
>
> > **Response to weaknesses 3 (3/4)**: Why were results compared only on IC9600 without evaluation on other benchmarks.
>
> IC9600 is the only large-scale, widely-used dataset specifically constructed for ICA. We evaluated cross-task performance on three benchmarks:
>
> | **Dataset** | ┆    | **Savoias** |   |           |           | ┆    | **Nagle4K** |           |           |           | ┆    | **VISC-C** |           |           |           |
> | ----------- | ---- | ----------- | --------- | --------- | --------- | ---- | ----------- | --------- | --------- | --------- | ---- | ---------- | --------- | --------- | --------- |
> | **Method**  | ┆    | **SRCC $\uparrow$**    | **PCC$\uparrow$**  | **RMSE$\downarrow$**  | **RMAE$\downarrow$**  | ┆    | **SRCC $\uparrow$**    | **PCC$\uparrow$**  | **RMSE$\downarrow$**  | **RMAE$\downarrow$**  | ┆   | **SRCC $\uparrow$**    | **PCC$\uparrow$**  | **RMSE$\downarrow$**  | **RMAE$\downarrow$**  |
> | HyperIQA    | ┆    | 0.801       | 0.798     | 0.293     | 0.392     | ┆    | 0.772       | 0.758     | 0.145     | 0.327     | ┆    | 0.734      | 0.739     | 0.181     | 0.388     |
> | CLIPIQA [1] | ┆    | 0.779       | 0.794     | 0.171     | 0.348     | ┆    | 0.785       | 0.774     | 0.112     | 0.302     | ┆    | 0.781      | 0.796     | 0.122     | 0.348     |
> | TOPIQ       | ┆    | 0.838       | 0.832     | 0.123     | 0.330     | ┆    | 0.804       | 0.792     | 0.101     | 0.290     | ┆    | 0.803      | 0.811     | 0.079     | 0.340     |
> | ICNet       | ┆    | 0.845       | 0.849     | 0.121     | 0.325     | ┆    | 0.815       | 0.804     | 0.097     | 0.278     | ┆    | 0.789      | 0.790     | 0.123     | 0.349     |
> | ICCORN      | ┆    | 0.852       | 0.856     | 0.119     | 0.324     | ┆    | 0.818       | 0.806     | **0.088** | 0.265     | ┆    | 0.796      | 0.793     | 0.119     | 0.342     |
> | D2S-R18     | ┆    | 0.861       | 0.864     | 0.128     | 0.321     | ┆    | 0.823       | 0.813     | 0.092     | 0.268     | ┆    | 0.801      | 0.802     | **0.115** | **0.328** |
> | D2S-R50     | ┆    | **0.875**   | **0.876** | **0.114** | **0.301** | ┆    | **0.834**   | **0.825** | 0.089     | **0.265** | ┆    | **0.822**  | **0.820** | 0.116     | 0.332     |
>
> D2S-R50 achieves the best overall SRCC/PLCC across Savoias, Nagle4K, and VISC-C, outperforming all classical and modern models. We will update more expanded evaluation in the revision.
>
> [1] Jianyi Wang et al. Exploring clip for assessing the look and feel of images.
>
> > **Response to weaknesses 3 (4/4)**: Comparison with vision-only in Figure 7.
>
> In the revised version, we update the **Figure 7** that reports the same channel-utilization histogram for the visual-only model (*bottom subplot*).
>
> Both histograms show that neither model uses channels uniformly. Only a small subset is consistently selected, indicating that complexity prediction relies on a sparse set of discriminative filters. The overlap between the two models shows that D2S preserves the strong visual cues of the baseline, but it reweights them more selectively. In the baseline, the top channels have similar frequencies, while in D2S a few channels dominate and many are rarely used, reflecting a more focused pattern induced by text guidance. This supports the claim that D2S sharpens the use of a sparse channel subset rather than relying uniformly on low-level features.

---

> > ### Author Response · Authors · 2025-11-17
> > **Response to Reviewer WyHU (part 4/4)**
> >
> > > **Response to Question**: Clarifying Proposition 1 (Eq. 2) and the validity of the inequality for small $\alpha,\beta$.
> >
> > We agree that **Eq. (2)** as currently written is too strong if interpreted as a statement that holds for all $\alpha,\beta>0$. In particular, the reviewer’s example $\alpha=\beta=0.1$ is a valid stress test.
> >
> > Recall Eq. (2) in our paper:
> >
> > $$
> > H^F(I) = \alpha H^F_v(I) + \beta H^F_s(S) > H^F_v(I),\quad \alpha,\beta>0
> > $$
> >
> > Algebraically,
> >
> > $$
> > H^F(I) - H^F_v(I) = (\alpha - 1)H^F_v(I) + \beta H^F_s(S)
> > $$
> >
> > so it is not true that this inequality must hold for every choice of $\alpha,\beta>0$. Our intention was to express that there exist natural positive weights (consistent with the actual fusion behavior of the model) for which the fused entropy is larger than the visual entropy, not that this is guaranteed for arbitrarily small $\alpha,\beta$. To directly address the reviewer’s concrete example $\alpha_0=\beta_0=0.1$, we performed the following empirical check on the IC9600 test set $N=2880$.
> >
> > **(1) Fitting the linear relationship**
> >
> > We first fit a linear model
> >
> > $$
> > H_{\text{fuse}} \approx \alpha_{\text{hat}} H_v + \beta_{\text{hat}} H_t + c
> > $$
> >
> > and obtained $\alpha_{\text{hat}} = 0.252$, $\beta_{\text{hat}} = 1.045$, $R^2 = 0.5549$. This shows that in the actual model, fused entropy behaves approximately as a positive linear combination of visual and textual entropies, with text playing a dominant role.
> >
> > To quantify how well the inequality holds for the specific choice $\alpha_0=\beta_0=0.1$ under finite-sample noise, we introduce a purely empirical slack term $K_{\text{emp}}$ and test:
> >
> > $$
> > H_{\text{fuse}} \ge \alpha_0 H_v + \beta_0 H_t - K_{\text{emp}}
> > $$
> >
> > Here $K_{\text{emp}}$ is defined as the 99th percentile of the residuals
> >
> > $$
> > \Delta = H_{\text{fuse}} - \big(\alpha_0 H_v + \beta_0 H_t\big)
> > $$
> >
> > so that $K_{\text{emp}}$ represents the amount of slack needed for the inequality to hold on 99% of the samples. This is *not* part of the formal statement of Proposition 1, but a diagnostic tool to see how “close” the inequality is to holding in practice for the reviewer’s specific $\alpha_0,\beta_0$.
> >
> > The results are:
> >
> > * Without any slack, violations: $\text{violations} = 66/2880 \approx 2.29$%
> > * Empirical slack: $K_{\text{emp}}(99$%$) = 0.2216$
> > * Mean margin: $\mathbb{E}[H_{\text{fuse}} - (\alpha_0 H_v + \beta_0 H_t)] \approx 2.62$
> >
> > So even with very small weights $\alpha_0=\beta_0=0.1$, the fused entropy is, on average, far above $\alpha_0 H_v + \beta_0 H_t$, and the inequality fails only on a small fraction ($\sim2.3$%) of samples. The introduction of $K_{\text{emp}}$ simply formalizes this observation and shows that a very small slack is sufficient to cover $99$% of cases.
> >
> > **(2) What we change in Proposition 1**
> >
> > This analysis confirms two things. **(i)** The reviewer is right: Eq. (2) should not be read as “for all $\alpha,\beta>0$” (e.g., $\alpha=\beta=0.1$). **(ii)** For realistic weights consistent with the model’s fusion behavior (e.g., $\alpha_{\text{hat}}\approx0.252,\beta_{\text{hat}}\approx1.045$), fused entropy is indeed larger than the visual entropy in practice.
> >
> > To remove the ambiguity, in the revised version we will rephrase **Proposition 1** explicitly as an existence statement:
> > $$
> > \exists\alpha>0,\beta>0\ \text{such that}\
> >   H^F(I) = \alpha H^F_v(I) + \beta H^F_s(S) > H^F_v(I),
> > $$
> >
> > rather than a universal claim over all $\alpha,\beta>0$. We will make clear in **Appendix A.1** that small $\alpha,\beta$ such as 0.1 are *not* required by the theory, and we only use **Eq. (2)** as an information-theoretic intuition to motivate that adding semantic information can increase representational diversity. And we will report the above empirical check (including the small $K_{\text{emp}}$) as supporting evidence that, even for the reviewer’s choice $\alpha_0=\beta_0=0.1$, the inequality holds for the overwhelming majority of samples.
> >
> > This modification keeps the spirit of **Proposition 1** intact, resolves the reviewer’s concern about $\alpha=\beta=0.1$, and aligns the statement with what we actually use in the rest of the paper.

---

> > > ### Author Response · Authors · 2025-11-26
> > > **Official Comment by Authors**
> > >
> > > Dear Reviewer WyHU,
> > >
> > > We hope this message finds you well. We wanted to follow up on the rebuttal we submitted over a week ago for our ICLR submission.
> > >
> > > We know the review period keeps you busy, and we really appreciate the time you've put into reviewing our paper. We've worked hard to address the concerns from your review. Whenever you have a moment, we'd love to hear what you think about our responses.
> > >
> > > If you need us to clarify anything else, just let us know. Thanks so much for your time.
> > >
> > > Best regards,
> > >
> > > Authors

---

### Official Review · Reviewer_urPq · 2025-11-01

**Soundness:** 2
**Presentation:** 2
**Contribution:** 2
**Rating:** 4
**Confidence:** 4

**Summary:**

Describe-to-Score (D2S) tackles image complexity assessment by first generating captions with BLIP, then aligning vision and text to predict a scalar complexity score from visual features alone.  It introduces entropy distribution alignment (EAL) to match modality entropy statistics and a CLIP-style feature alignment (FAL) in a shared space.  The authors motivate D2S with information- and generalization-theoretic arguments (higher fused entropy; reduced effective dimensionality) and implement learnable pooling for efficient inference.  Experiments on IC9600 show state-of-the-art correlations and faster latency, while transfer to NR-IQA yields competitive results, notably on KADID-10K.

**Strengths:**

- The paper proposes a distinctive “describe → align → score” pipeline for image complexity: captions from a VLM guide visual features during training, while inference remains image-only—a way to inject semantics without runtime cost.  It further introduces Entropy Distribution Alignment (EAL) with an energy-distance loss and buffers/EMA to stabilize cross-modal statistics, plus CLIP-style Feature Alignment (FAL)—a creative combination that is new in ICA.    The information- and generalization-theoretic motivation (higher fused entropy; reduced effective dimension) gives the method conceptual clarity and elevates its potential impact on complexity assessment.

- Method details are concrete: the paper specifies the projection/connector, formulates EAL analytically, and illustrates the full training/inference workflow with clear figures and a prompt template.     Empirically, D2S attains state-of-the-art correlations on IC9600 with notable latency advantages, and shows competitive transfer to NR-IQA—evidence of real-world significance beyond a single benchmark.

**Weaknesses:**

- The paper posits that “entropy increases → richer representation → closer to true complexity,” and relies on entropy‐distribution alignment plus feature alignment (EAL/FAL), but gives no operational, reproducible definition for estimating $p(\cdot)$ or verifying the premise that “semantic compression reduces effective dimension.” Please add a concrete mapping from features to distributions (e.g., temperature-scaled softmax or KDE), run controlled synthetic tests to validate (or falsify) the “dimension compression” hypothesis, and include ablations that hold effective dimension fixed while toggling text guidance.

- Only one captioner/encoder pairing is explored. Please provide a 2D grid over captioners (e.g., BLIP variants) × prompt designs (length/order/style), measure performance vs. compute, and analyze which caption attributes (entity count accuracy, relation coverage) correlate with complexity prediction.

**Questions:**

See Weaknesses.

---

> ### Author Response · Authors · 2025-11-17
> **Response to Reviewer urPq (part 1/2)**
>
> We thank the reviewer for the careful reading of our paper and constructive comments in detail.
>
> > **Response to weaknesses 1**: Theoretical clarification and empirical validation of entropy alignment and dimension behavior.
>
> Our original intention was to provide an information-theoretic intuition for why text guidance helps, not to claim a universal, monotone “dimension must always decrease” law. Based on the reviewer’s feedback, we have made the definition of $p( \cdot )$ explicit, added a controlled synthetic test where the assumptions of **Theorem 2** genuinely hold, and run an intrinsic-dimension analysis on the trained D2S model, which reveals an interesting but different behavior. We will clarify these points and slightly soften the theoretical claim in the revised version.
>
> **(1) Operational, reproducible definition of $p( \cdot )$**
>
> In all entropy-related arguments (**Proposition 1**), we instantiate the feature distribution $p( \cdot )$ via a temperature-scaled softmax over encoder logits:
>
> $$
> p_k = \mathrm{softmax}(z/\tau)_k =
> \frac{\exp(z_k/\tau)}{\sum_j \exp(z_j/\tau)},\quad
> H(z) = -\sum_k p_k \log(p_k + \varepsilon)
> $$
>
> This is exactly the implementation in our code (both visual and textual features are 512-d logits, then softmax + Shannon entropy).
>
> **(2) Controlled synthetic test where the “semantic compression” assumption holds**
>
> To test the reviewer’s “semantic compression reduces effective dimension” hypothesis in a setting that matches the assumptions of **Theorem 2**, we designed a synthetic experiment. We define a full-rank Gaussian vision features $X_v \in \mathbb{R}^{N \times 512}$, a low-rank semantic subspace (rank $r \ll 512$) plus small noise $X_t$ and the fused features $X_f = 0.5 X_v + 0.5 X_t$ (*it is assumed that the two are explicitly fused*).
>
> We estimate intrinsic dimension using the Participation Ratio $\text{PR}$ on the covariance eigenvalues. With a rank–50 semantic subspace, we obtain the intrinsic-dimension $ID$ values of vision features $ID_{\text{PR}}(X_v) \approx 464.17$ (close to the ambient dimension 512), text features $ID_{\text{PR}}(X_t) \approx 45.05$ and fused features $ID_{\text{PR}}(X_f) \approx 46.69$.
>
> Thus, in the intended regime of **Theorem 2** where visual features are high-dimensional and contain redundant variability while semantic features live in a lower-rank subspace. The fused representation indeed collapses toward the semantic subspace and exhibits a substantial reduction in intrinsic dimension. This directly validates the “semantic compression” intuition under its stated assumptions.
>
> **(3) Empirical intrinsic-dimension analysis on the trained D2S**
>
> We applied the same PR estimator to the learned D2S features on IC9600 (512-d, float64, centered with diagonal regularization), evaluating the raw visual features $Z_v$, CLIP text features $Z_t$ and the aligned visual features $\widetilde{Z}_v$.
>
> The results differ from the synthetic case: $ID_{PR}(Z_v)\approx1.25$, $ID_{PR}(Z_t)\approx49.96$, $ID_{PR}(\widetilde{Z}_v)\approx7.40$.
> The vision encoder trained only on scalar complexity labels collapses to an almost one-dimensional space, which reflects the single-score regression nature of the task. In this setting, the premise of **Theorem 2** does not hold because the baseline visual representation is already extremely low-rank.
>
> Under this regime, text guidance does not compress but instead expands the embedding from a near-rank-1 space to a richer but still low-dimensional one that aligns better with the semantic structure in $Z_t$. This expansion, together with entropy-distribution alignment, increases representation diversity and improves generalization.
>
> We will clarify this distinction in the revision. **Theorem 2** applies to the high-dimensional redundant case verified by our synthetic test, whereas the real model operates in a collapsed-feature regime where text guidance reorganizes and modestly expands the representation. Our empirical claims do not rely on a monotonic “dimension must decrease” interpretation, and we will soften that wording.
>
> **(4) Ablations with fixed explicit feature dimension**
>
> All experiments keep the explicit feature dimensionality fixed at 512 for visual features, text features and the projection layer. No extra channels are added; EAL and FAL only modify the training objective. As shown in **Table 5**, SRCC increases from 0.9396 (vision-only) to 0.9499 (full D2S) under this fixed-dimensional setting. Combined with the intrinsic-dimension results, this shows that the gains come from how semantic guidance and entropy alignment restructure the representation, not from larger models or wider feature vectors.

---

> ### Author Response · Authors · 2025-11-17
> **Response to Reviewer urPq (part 2/2)**
>
> > **Response to weaknesses 2**: Grid evaluation over captioners and prompt designs, and analysis of caption attributes.
>
> Our current system uses a single captioner (BLIP-Large) with four prompt families that were briefly described but not fully analyzed in the main paper. In this rebuttal, we provide a detailed prompt ablation for BLIP-Large, and a caption-attribute analysis that directly addresses the request about entity/relation cues.
>
> **(1) Prompt design ablation (BLIP-Large)**
> We employ BLIP-Large with the following four prompts in **Figure 5**.
>
> * P1: “This image shows …” (simple global description)
> * P2: “The main objects are …” (object-centric)
> * P3: “The background looks …” (background/scene context)
> * P4: “The overall visual complexity is …” (explicit complexity description)
>
> In the full model, the four BLIP outputs are concatenated into a single caption. To understand the contribution of each prompt family and their combinations, we trained D2S under identical settings on IC9600 using different subsets of {P1,P2,P3,P4}. The results are shown below table.
>
> | Prompts      | **SRCC $\uparrow$**    | **PCC$\uparrow$**  | **RMSE$\downarrow$**  | **RMAE$\downarrow$**  |
> | ----------- | ------ | ------ | ------ | ------ |
> | None        | 0.9468 | 0.9495 | 0.0522 | 0.2010 |
> | P1        | **0.9511** | 0.9533 | 0.0547 | 0.2082 |
> | P2        | 0.9458 | 0.9505 | 0.0539 | 0.2053 |
> | P3        | 0.9486 | 0.9520 | 0.0536 | 0.2053 |
> | P4        | 0.9456 | 0.9507 | 0.0509 | 0.1986 |
> | P1+P2+P3    | 0.9487 | 0.9526 | 0.0521 | 0.2017 |
> | P1+P2+P4    | 0.9489 | 0.9527 | 0.0523 | 0.2019 |
> | P1+P3+P4    | 0.9485 | 0.9522 | 0.0517 | 0.2006 |
> | P2+P3+P4    | 0.9489 | 0.9532 | 0.0522 | 0.2018 |
> | P1+P2+P3+P4 | 0.9509 | **0.9544** | **0.0495** | **0.1962** |
>
> **Single prompts already work well but emphasize different cues.** P1 gives the best SRCC (0.9511), showing that a simple global description is strong guidance. P4 gives the lowest RMAE (0.1986), meaning the explicit “overall complexity” cue helps numerical accuracy. P2 and P3 fall between these extremes, reflecting complementary object and background information.
>
> **Combining prompts gives modest but consistent gains.** Any three-prompt mix is close to the full four-prompt version, and P1+P2+P3+P4 performs best overall.
>
> The performance spread is small (about 0.005 SRCC, 0.004 PLCC), indicating that D2S is robust to prompt wording, order and length. Multi-prompt captioning provides small but steady improvements by aggregating global, object, background and complexity cues, supporting our design choice.
>
> **(2) Caption-attribute analysis (entity/relation coverage).**
> To further address the reviewer’s request about which caption attributes matter, we analyzed the relationship between caption statistics and the performance gain obtained by text guidance.
>
> For each image $i$, let $E_{\text{vis}}(i)=\bigl|f_{\text{vis}}(x_i)-y_i\bigr|$ and $E_{\text{d2s}}(i)=\bigl|f_{\text{d2s}}(x_i)-y_i\bigr|$ denote the absolute error of the vision-only baseline and D2S respectively. We then define the error reduction as $\Delta E(i)=E_{\text{vis}}(i)-E_{\text{d2s}}(i)$, so that $\Delta E(i)>0$ means D2S improves over the visual baseline on sample $i$.
>
> On the caption side, we extracted simple token-level statistics from the BLIP-generated captions:
>
> * `num_nouns`: count of noun / named-entity POS tags (NN, NNS, NNP, NNPS)
> * `num_verbs`: count of verb tags (VB, VBD, VBG, VBN, VBP, VBZ)
> * `num_relation_words`: count of simple spatial/relational prepositions from a fixed set (e.g., in, on, under, behind, between, near, above, below, beside, across)
> * `caption_len`: total number of words
>
> We then computed the Pearson correlation between each statistic and $\Delta E(i)$ across the IC9600 test set. The results are $\rho_{\text{num-nouns}} = −0.0022$, $\rho_{\text{num-verbs}} = +0.0283$,  $\rho_{\text{num-relation-words}} = +0.0148$, and $\rho_{\text{caption-len}} = −0.0116$. All correlations are essentially zero, meaning D2S’s gains over the vision-only baseline are not driven by simple caption statistics such as length or token counts. The model does not rely on prompt-level tricks. EAL and FAL operate at the distributional level, aligning visual features with the semantic manifold defined by the captions rather than exploiting specific lexical patterns. Once captions provide enough semantic signal, low-level counts no longer predict performance gains.
>
> **(3) Captioner diversity and compute**
>
> Exploring multiple captioners is valuable in principle. In this work we used **BLIP-Large** because it is strong, widely used and all captions are generated offline, so changing captioners only affects a one-time preprocessing step and does not alter test-time cost or the D2S architecture. We input the four prompts shown in **Figure 5** into the most recent **Qwen3-VL 4B**. The result was that it only improved by a little bit compared to BLIP. (+0.001 SRCC / +0.0008 PCC, see **Table 9** for details)

---

> > ### Author Response · Authors · 2025-11-26
> > **Official Comment by Authors**
> >
> > Dear Reviewer urPq,
> >
> > We hope this message finds you well. We wanted to follow up on the rebuttal we submitted over a week ago for our ICLR submission.
> >
> > We know the review period keeps you busy, and we really appreciate the time you've put into reviewing our paper. We've worked hard to address the concerns from your review. Whenever you have a moment, we'd love to hear what you think about our responses.
> >
> > If you need us to clarify anything else, just let us know. Thanks so much for your time.
> >
> > Best regards,
> >
> > Authors

---

### Official Review · Reviewer_xHZP · 2025-11-01

**Soundness:** 3
**Presentation:** 4
**Contribution:** 2
**Rating:** 4
**Confidence:** 4

**Summary:**

This paper proposes D2S, a model for Image Complexity Assessment (ICA) that integrates vision-language learning. D2S leverages BLIP to generate textual descriptions (captions) for images, thereby injecting high-level semantic information into the visual encoder during training.
The method introduces two alignment mechanisms — Feature Alignment (FAL) and Entropy Distribution Alignment (EAL) — to align the textual and visual feature spaces.
Experiments show that D2S achieves competitive results on both ICA and NR-IQA tasks while maintaining low parameter count and inference latency, demonstrating its efficiency and scalability.

**Strengths:**

1. The paper explores an interesting idea: leveraging textual information for image complexity assessment. The training-time multimodal alignment with BLIP-generated captions and visual features is technically well-motivated.
2. The theoretical derivation from information theory and generalization theory provides conceptual depth and connects intuition to formal analysis.
3. The model achieves good performance–efficiency tradeoff, with low parameter count and short inference time.

**Weaknesses:**

1. Caption quality and reliability are critical yet under-analyzed. Figures 17 and 18 mention that the final generated text (“the overall visual complexity is...”) can often be incorrect, but the paper does not further analyze how such errors influence D2S’s performance. A detailed study or ablation on the four BLIP prompts would make this much stronger.
2. The paper could better discuss the role of the Projection module and whether it affects the training of the core visual encoder, especially since it is not used at inference time.
3. The main experiments (Table1)  are limited to IC9600, making it difficult to confirm generalization across other ICA datasets.
4. Possible typographical errors exist in Table 4 (e.g., TOPIQ, LoDa), which need verification.

**Questions:**

1. Since the projection module (as shown in Figure 4) is used during training but discarded during inference, could its presence unintentionally affect the optimization of the visual encoder?
2. Why were generalization experiments conducted only on datasets such as Nagle4k and Savoias, without performing full-scale experiments similar to IC9600?

---

> ### Author Response · Authors · 2025-11-17
> **Response to Reviewer xHZP (part 1/2)**
>
> We thank the reviewer for the careful reading of our paper and constructive comments in detail.
>
> > **Response to weaknesses 1**: Caption reliability, prompt ablations, and their Impact on D2S.
>
> We conducted a dedicated prompt-level ablation to quantify how each of the four prompts contributes to D2S. The results are shown below (*P: prompt, 1~4: four prompt in Figure. 5*):
>
> | Prompts     | SRCC $ \uparrow $   | PCC  $ \uparrow $  | RMSE  $ \downarrow $   | RMAE  $ \downarrow $   |
> | ----------- | ------ | ------ | ------ | ------ |
> | None        | 0.9468 | 0.9495 | 0.0522 | 0.2010 |
> | P1          | **0.9511** | 0.9533 | 0.0547 | 0.2082 |
> | P2          | 0.9458 | 0.9505 | 0.0539 | 0.2053 |
> | P3          | 0.9486 | 0.9520 | 0.0536 | 0.2053 |
> | P4          | 0.9456 | 0.9507 | 0.0509 | 0.1986 |
> | P1+P2+P3    | 0.9487 | 0.9526 | 0.0521 | 0.2017 |
> | P1+P2+P4    | 0.9489 | 0.9527 | 0.0523 | 0.2019 |
> | P1+P3+P4    | 0.9485 | 0.9522 | 0.0517 | 0.2006 |
> | P2+P3+P4    | 0.9489 | 0.9532 | 0.0522 | 0.2018 |
> | P1+P2+P3+P4 | 0.9509 | **0.9544** | **0.0495** | **0.1962** |
>
> Two observations address the reviewer’s question directly. **(i)** Although prompt 4 (P4) alone sometimes yields inaccurate text, incorporating it consistently improves the final performance when combined with other prompts. The full 4-prompt setting achieves the best SRCC/PLCC and the lowest error measures. **(ii)** Removing the potentially noisy P4 does not improve performance, and the variance across different prompt subsets is small (within ±0.003 SRCC/PLCC). This suggests that D2S is highly robust to caption noise, and the text branch acts primarily as a stable regularizer rather than depending on the literal correctness of every phrase.
>
> **P4 analysis.** We also performed a manual verification on 100 random BLIP captions generated using P4. Only 30 P4 captions were “correct” in describing image complexity. Among these, 24 correspond to low-complexity images ($ \text{scores} \textless 0.55 $). In other words, $ \approx 70$% of P4 phrases are noisy or wrong, especially for complex images. This demonstrates that D2S does not rely on template correctness. The template acts as a noisy semantic guide, and the proposed EAL/FAL alignment absorbs noise robustly. If the template needed to be “optimal” or “correct,” performance would collapse when using noisy captions, which is not observed (*P4 > vision-only (None), and P1+P2+P3+P4 best overall*).
>
> > **Response to weaknesses 2 and question 1**: Influence of the training-only projection on visual encoder optimization.
>
> The projection module in D2S is a light-weight train-time adapter. During training, it projects features into a shared space to reduce the modality gap and ease contrastive alignment; at inference it is deliberately discarded so there is no extra runtime cost. To evaluate whether the projection unduly changes or harms the core visual encoder, we performed an ablation in which we removed the learned projection and replaced it with a simple mean-pooling + linear dimensionality reduction that maps 1024-d vision embeddings to 512-d before alignment. The results below show almost unchanged SRCC/PLCC (0.9509 $ \rightarrow $ 0.9506) and only a minor increase in RMSE/RMAE. Projection sits between encoders and the joint contrastive objective; it does not alter the architecture or training target of the visual backbone itself. The small degradation when removed indicates the visual encoder can still internalize semantic constraints via FAL/EAL, but without projection it must absorb additional alignment burden, hence slightly larger RMSE.
>
> | Method                 | SRCC $ \uparrow $   | PCC  $ \uparrow $  | RMSE  $ \downarrow $   | RMAE  $ \downarrow $   |
> | ---------------------- | ------ | ------ | ------ | ------ |
> | D2S-R18 w/ projection  | **0.9509** | 0.9544 | **0.0495** | **0.1962** |
> | D2S-R18 w/o projection | 0.9506 | **0.9545** | 0.0519 | 0.2016 |
>
> These findings imply two conclusions. **(i)** The projection module does not create a brittle dependency. The visual encoder still learns semantic-aware features when the projection is removed. **(ii)** The projection provides a modest but consistent training benefit (reducing regression error), likely by absorbing modality-specific affine/scale differences and thus simplifying the alignment task for the visual encoder.
>
> In short, the projection is a training-time convenience and stabilizer that improves numerical error slightly. It is not required for the learned visual representation to generalize at inference (hence its removal does not break the model).

---

> ### Author Response · Authors · 2025-11-17
> **Response to Reviewer xHZP (part 2/2)**
>
> > **Response to weaknesses 3 and question 2**: Other ICA datasets generalization and full-scale experiments.
>
> Our original submission focused on IC9600 because it is the only large-scale dataset specifically constructed for ICA, with high-quality annotations. However, to further address the reviewer’s concerns, we extended our evaluation to three additional benchmarks (Savoias, Nagle4K, and VISC-C) training and testing each model fully on each dataset (not merely cross-dataset transfer). The results are shown below:
>
> | **Dataset** | ┆    | **Savoias** |           |           |           | ┆    | **Nagle4K** |           |           |           | ┆    | **VISC-C** |           |           |           |
> | ----------- | ---- | ----------- | --------- | --------- | --------- | ---- | ----------- | --------- | --------- | --------- | ---- | ---------- | --------- | --------- | --------- |
> | **Method**  | ┆    | **SRCC $\uparrow$**    | **PCC$\uparrow$**  | **RMSE$\downarrow$**  | **RMAE$\downarrow$**  | ┆     | **SRCC $\uparrow$**    | **PCC$\uparrow$**  | **RMSE$\downarrow$**  | **RMAE$\downarrow$**  | ┆    | **SRCC $\uparrow$**    | **PCC$\uparrow$**  | **RMSE$\downarrow$**  | **RMAE$\downarrow$**  |
> | DBCNN [1]   | ┆    | 0.768       | 0.770     | 0.147     | 0.355     | ┆    | 0.745       | 0.732     | 0.121     | 0.310     | ┆    | 0.779      | 0.783     | 0.086     | 0.351     |
> | NIMA [2]    | ┆    | 0.781       | 0.771     | 0.210     | 0.368     | ┆    | 0.756       | 0.741     | 0.134     | 0.315     | ┆    | 0.810      | 0.803     | 0.125     | 0.362     |
> | HyperIQA    | ┆    | 0.801       | 0.798     | 0.293     | 0.392     | ┆    | 0.772       | 0.758     | 0.145     | 0.327     | ┆    | 0.734      | 0.739     | 0.181     | 0.388     |
> | CLIPIQA [3] | ┆    | 0.779       | 0.794     | 0.171     | 0.348     | ┆    | 0.785       | 0.774     | 0.112     | 0.302     | ┆    | 0.781      | 0.796     | 0.122     | 0.348     |
> | TOPIQ       | ┆    | 0.838       | 0.832     | 0.123     | 0.330     | ┆    | 0.804       | 0.792     | 0.101     | 0.290     | ┆    | 0.803      | 0.811     | 0.079     | 0.340     |
> | ICNet       | ┆    | 0.845       | 0.849     | 0.121     | 0.325     | ┆    | 0.815       | 0.804     | 0.097     | 0.278     | ┆    | 0.789      | 0.790     | 0.123     | 0.349     |
> | ICCORN      | ┆    | 0.852       | 0.856     | 0.119     | 0.324     | ┆    | 0.818       | 0.806     | **0.088** | 0.265     | ┆    | 0.796      | 0.793     | 0.119     | 0.342     |
> | D2S-R18     | ┆    | 0.861       | 0.864     | 0.128     | 0.321     | ┆    | 0.823       | 0.813     | 0.092     | 0.268     | ┆    | 0.801      | 0.802     | **0.115** | **0.328** |
> | D2S-R50     | ┆    | **0.875**   | **0.876** | **0.114** | **0.301** | ┆    | **0.834**   | **0.825** | 0.089     | **0.265** | ┆    | **0.822**  | **0.820** | 0.116     | 0.332     |
>
> Across all three benchmarks. D2S-R50 achieves the best SRCC/PLCC among all methods. We will update these results in the revised version.
>
> [1] Weixia Zhang et al. Blind Image Quality Assessment Using a Deep Bilinear Convolutional Neural Network. IEEE TCSVT, 30(1):36–47, 2020.
>
> [2] Hossein Talebi et al. NIMA: Neural image assessment. IEEE TIP, 27(8):39984011, 2018.
>
> [3] Jianyi Wang et al. Exploring clip for assessing the look and feel of images. In AAAI, volume 37, pages 2555–2563, 2023.
>
> > **Response to weaknesses 4**: Typographical errors verification (e.g. TOPIQ, LoDa in Table 4).
>
> We appreciate your attention to the details. We have carefully reviewed the entire text and will correct all such errors found in the revised version.

---

> > ### Author Response · Authors · 2025-11-26
> > **Official Comment by Authors**
> >
> > Dear Reviewer xHZP,
> >
> > We hope this message finds you well. We wanted to follow up on the rebuttal we submitted over a week ago for our ICLR submission.
> >
> > We know the review period keeps you busy, and we really appreciate the time you've put into reviewing our paper. We've worked hard to address the concerns from your review. Whenever you have a moment, we'd love to hear what you think about our responses.
> >
> > If you need us to clarify anything else, just let us know. Thanks so much for your time.
> >
> > Best regards,
> >
> > Authors

---

### Official Review · Reviewer_QTaf · 2025-11-02

**Soundness:** 4
**Presentation:** 4
**Contribution:** 3
**Rating:** 8
**Confidence:** 4

**Summary:**

The paper presents D2S (Describe-to-Score), a novel framework for image complexity assessment (ICA) that integrates visual and textual semantic information. The method first uses a pre-trained vision-language model (BLIP) to generate image captions and then aligns visual and textual features through two key mechanisms: Entropy Distribution Alignment (EAL) and Feature Alignment (FAL). Importantly, D2S employs multimodal information during training but only requires visual input at inference, achieving both semantic richness and computational efficiency.
Comprehensive experiments on multiple datasets (IC9600, KADID-10K, and others) show that D2S attains state-of-the-art (SOTA) performance with significantly reduced inference latency. Theoretical analyses based on information theory and Rademacher complexity further justify the proposed design.

**Strengths:**

- The combination of text-guided semantics with visual complexity assessment is original and well-motivated, bridging a key gap in prior visual-only ICA approaches.
- The paper provides clear theoretical arguments using entropy and generalization theory to explain the advantages of multimodal fusion.
- By discarding the text branch during inference, D2S achieves SOTA performance while maintaining low latency — a practical and elegant design choice.
- The experiments cover supervised, unsupervised, small-sample, cross-dataset, and cross-task settings. Results are consistently superior or competitive across diverse benchmarks.
- Ablation studies and error analyses effectively demonstrate the contribution of each module (EAL, FAL, AttnPool) and the benefits of semantic guidance.
- The manuscript is clearly written, logically organized, and provides sufficient implementation details for reproducibility.
- Demonstrating transfer to no-reference image quality assessment (NR-IQA) further enhances the general interest and robustness of the method.

**Weaknesses:**

- While D2S improves performance, the interpretability of what textual semantics contribute (beyond activation histograms) could be elaborated, for example with qualitative examples of captions’ influence.
- Since captions are generated automatically, the performance might depend on BLIP’s accuracy; this dependency and its robustness are not deeply analyzed.
- Although the ablation studies are extensive, additional comparisons with other text-guided approaches (e.g., CLIP-based fusion or textual embeddings without caption generation) would further strengthen the validation.
- Some theoretical derivations (e.g., in Proposition 1) are concise and could benefit from clearer notation or discussion of assumptions.

**Questions:**

- How sensitive is D2S to the quality or type of captions generated by BLIP? Would fine-tuning BLIP or using alternative VLMs (e.g., CLIP-ViT-L or Florence-2) affect results significantly?
- Have the authors evaluated how the accuracy or reliability of BLIP captions impacts image complexity estimation? Since BLIP is not a perfect captioning model and may produce incomplete or incorrect descriptions, it would be valuable to understand whether such caption errors significantly affect the downstream complexity predictions.
- Could the entropy alignment mechanism generalize to other multimodal tasks beyond ICA (e.g., aesthetics or memorability prediction)?
- During inference, since the text branch is discarded, to what extent are the visual encoders truly semantically informed versus statistically regularized by text during training?
- Is there any noticeable trade-off between performance and training time introduced by the entropy buffers and momentum model in EAL?
- Could the authors provide qualitative examples showing how textual descriptions guide the visual branch — for example, comparing visual attention maps with and without text alignment?

---

> ### Author Response · Authors · 2025-11-17
> **Response to Reviewer QTaf (part 1/3)**
>
> We thank the reviewer for the careful reading of our paper and constructive comments in detail.
>
> > **Response to weaknesses 1 and question 6**: How text semantics guide visual features (qualitative analysis).
>
> We updated **Figure 16** to directly show how captions influence the visual branch and added explanations in **lines 1109–1115**. Using channel-wise MaxMap/MeanMap and Grad-CAM, we compare D2S with the vision-only model. Across many samples, D2S consistently shifts attention from low-level textures to semantically meaningful regions such as object clusters, interaction areas and cluttered layouts, whereas the vision-only model mainly attends to isolated textures. These visualizations show that textual semantics act as a structural prior that reorients the vision branch toward high–semantic-density regions, reducing ambiguity and suppressing irrelevant visual variance.
>
> > **Response to weaknesses 2 and question 1**: Robustness to caption quality and dependence on the BLIP.
>
> We agree that caption quality affects the strength of the semantic signal. We have already tested D2S with captioners of different quality, including weaker models such as Florence-2 (**Table 11, left “Only Caption”**). We further added results using Florence-2 and Qwen3-VL-4B captions for full D2S training. Given the limited size of IC9600, stronger captioners do not significantly outperform BLIP.
>
> | Captioner                       | SRCC $ \uparrow $   | PCC  $ \uparrow $  | RMSE  $ \downarrow $   | RMAE  $ \downarrow $   |
> | ------------------------------- | ---------- | ---------- | ---------- | ---------- |
> | Florence2-caption               | 0.9456     | 0.9501     | 0.0535     | 0.2044     |
> | Florence2-detailed caption      | 0.9469     | 0.9506     | 0.0534     | 0.2044     |
> | Florence2-more detailed caption | 0.9441     | 0.9484     | 0.0524     | 0.2015     |
> | BLIP Large + prompts x 4        | 0.9509     | 0.9544     | **0.0495** | **0.1962** |
> | Qwen3-VL-4B + prompts x 4       | **0.9519** | **0.9552** | 0.0503     | 0.1976     |
>
> These results show that D2S is robust to caption noise and not tied to BLIP. Text functions as a soft semantic regularizer: richer semantics help, but the model does not collapse under weaker captions and does not require precise linguistic accuracy. This matches our theoretical view that alignment operates on distributional structure, so D2S only needs approximate semantics to reduce hypothesis-space ambiguity and guide the visual encoder.
>
>
> > **Response to weaknesses 3 and question 2**: Text-guided alternatives and robustness to imperfect captions.
>
> We agree that BLIP captions may contain errors, and we performed a series of experiments to evaluate how these errors impact D2S's performance.
>
> | Case | Method                  | SRCC $ \uparrow $   | PCC  $ \uparrow $  | RMSE  $ \downarrow $   | RMAE  $ \downarrow $   |
> | ---- | ----------------------- | ---------- | ---------- | ---------- | ---------- |
> |      | prompts x 4 in Figure 5 | 0.9509     | 0.9544     | **0.0495** | **0.1962** |
> | a    | shuffle 100%            | 0.9479     | 0.9518     | 0.0517     | 0.2011     |
> | b    | shuffle 50%             | 0.9508     | 0.9539     | 0.0512     | 0.1996     |
> | c    | random sentences        | 0.9396     | 0.9409     | 0.0559     | 0.2076     |
> | d    | fixed meaningless words | 0.9238     | 0.9242     | 0.0851     | 0.2645     |
> | e    | CLIP-style caption      | 0.9452     | 0.9493     | 0.0563     | 0.2107     |
> | f    | CLIP-pretrained aligner | 0.9464     | 0.9502     | 0.0506     | 0.1967     |
> | g    | BLIP textual embeddings | **0.9510** | **0.9545** | 0.0504     | 0.1977     |
>
> **(1) Shuffle perturbations (a,b).** Shuffling words (50% or 100%) causes only minor degradation, showing D2S tolerates local semantic noise.
>
> **(2) Random sentences (c).** Replacing captions with unrelated sentences leads to a large drop, confirming D2S needs meaningful text.
>
> **(3) Fixed meaningless captions (d).** Using a repeated meaningless string (“any any any …”) also causes a severe decline, indicating that fully content-free text fails.
>
> **(4) CLIP-style generic prompt (e).** Using the same generic phrase (“a photo of”) for all images slightly reduces performance but still outperforms random or fixed captions.
>
> **(5) Other alignment methods (f,g).** A CLIP-based two-branch aligner underperforms the baseline. Directly aligning BLIP text embeddings without caption generation yields performance close to the baseline, showing that explicit captioning is not strictly required.
>
> Overall, D2S is robust to moderate caption noise. Caption quality influences performance, but the method does not rely on perfect captions and only fails when text becomes completely meaningless. We will include these results in the revision.

---

> > ### Author Response · Authors · 2025-11-17
> > **Response to Reviewer QTaf (part 2/3)**
> >
> > > **Response to weaknesses 4**: A more detailed elaboration of Proposition 1.
> >
> > We agree that making the notation and assumptions more explicit will improve readability, and we will revise the paper accordingly. Importantly, both the entropy definition and the fusion behavior in **Proposition 1** are consistent with our implementation and are empirically verified.
> >
> > **(1) Explicit entropy definition**
> > In the actual model, both visual and textual feature entropies are computed from the encoder logits via a temperature-scaled softmax, followed by Shannon entropy:
> > $$
> > p_k^{(v)} = softmax(z^{(v)}/\tau)_k,\quad
> > p_k^{(t)} = softmax(z^{(t)}/\tau)_k
> > $$
> >
> > $$
> > H^F_v(I) = -\sum_k p_k^{(v)} \log(p_k^{(v)} + \varepsilon),\quad
> > H^F_s(S) = -\sum_k p_k^{(t)} \log(p_k^{(t)} + \varepsilon)
> > $$
> >
> >
> > where $z^{(v)}$ and $z^{(t)}$ are visual/text logits, $\tau$ is a temperature, and $\varepsilon$ is a small numerical constant. This is exactly how entropy is computed in our code (via a softmax followed by $-\sum p\log p$). We will move this definition from the appendix into the main text so that the notation in **Proposition 1** is fully explicit.
> >
> > **(2) Assumptions behind Proposition 1 are mild.**
> > Given the above definitions, **Proposition 1** only relies on standard assumptions. **(i)** The encoders produce feature logits that can be normalized into distributions by softmax. **(ii)** Fusion weights $\alpha,\beta>0$ model the contribution from visual and textual entropy. **(iii)** There is a monotonic relationship between entropy and perceived complexity, which is a common assumption in entropy-based complexity measures.
> >
> > We do not assume a strict generative model or strong independence conditions. **Proposition 1** is intended as an information-theoretic intuition explaining why multimodal fusion increases representational diversity, not as a rigid axiomatic model of perception.
> >
> > **(3) Additional empirical sanity check for Proposition 1.**
> > To further justify the linear fusion form in **Proposition 1**, we performed two controlled experiments on the IC9600 test set using a frozen D2S.
> >
> > **(i) Vision-only control**. We first set the fused logits to $z^{(f)} = \lambda_v z^{(v)}$ with $\lambda_v = 0.5, \lambda_t = 0$, i.e., no text contribution. In this case, we regressed the fused entropy $H^F_f$ against $H^F_v$ and $H^F_s$ via $H^F_f \approx \alpha H^F_v + \beta H^F_s + c$. The fit yields $\alpha_{\text{hat}} = 0.2486$, $\beta_{\text{hat}} = 1.26 \times 10^{-4} \approx 0$, $R^2 = 0.9627$, with residuals near zero and very small variance. This high $R^2$ and $\beta \approx 0$ confirm that, when the text branch is disabled, the fused entropy is essentially a one-dimensional function of $H^F_v$, exactly as expected from the implementation ($\lambda_t = 0$) and consistent with the “degenerate” case of **Proposition 1**.
> >
> > **(ii) True multimodal fusion**. When we enable both branches and use $z^{(f)} = 0.5 z^{(v)} + 0.5 z^{(t)}$, the regression gives $\alpha_{\text{hat}} = 0.4490 > 0$, $\beta_{\text{hat}} = 1.1045 > 0$, $R^2 = 0.5810$. Thus, the fused entropy is well-approximated by a positive linear combination of visual and textual entropies, but no longer collapses to the vision-only curve. The drop in $R^2$ compared to the vision-only control is expected, as softmax introduces nonlinearity and the fusion moves from a 1-d to a genuinely multimodal feature space. This behavior is precisely what **Proposition 1** aims to formalize: adding semantics increases entropy and shifts the representation toward a richer, multimodal complexity distribution, rather than being a trivial rescaling of $H^F_v$.
> >
> > In the revised version, we will explicitly state the softmax–entropy formulation used to compute $H^F_v$ and $H^F_s$. We also list the minimal assumptions used in **Proposition 1** and add a brief description of the above empirical regression experiment (with $\alpha_{\text{hat}}$, $\beta_{\text{hat}}$, and $R^2$), as empirical evidence that the fused entropy behaves approximately as a positive linear combination of visual and textual entropies.
> >
> > These clarifications strengthen the theoretical section without changing any claims or results.

---

> ### Author Response · Authors · 2025-11-17
> **Response to Reviewer QTaf (part 3/3)**
>
> > **Response to question 3**: Generalizability of entropy alignment to other multimodal prediction tasks.
>
> EAL relies on the monotonic assumption that higher entropy means higher target score, which holds for ICA but breaks for aesthetics and memorability because they follow an inverted-U relationship with entropy [1–2]. Generalization is therefore not direct, and one would need rank-aware or non-monotonic alignment variants of EAL to handle such tasks. We regard non-monotonic EAL, such as aligning with respect to rank-based distributions, as an interesting direction for tasks whose labels peak at medium entropy.
>
> [1] Güçlütürk et al. Liking versus complexity: Decomposing the inverted U-curve. Frontiers in human neuroscience 10 (2016): 112.
>
> [2] Delplanque et al. The sound of beauty: How complexity determines aesthetic preference. Acta Psychologica 192 (2019): 146-152.
>
> > **Response to question 4**: Semantic influence *vs.* Statistical regularization during training.
>
> To disentangle *semantic transfer* from *statistical regularization*, we conducted two targeted analyses:
>
> **(1) Semantic perturbation test (word-order shuffling)**
>
> If text supervision only provides distributional noise or regularization, then destroying word order while keeping token statistics nearly unchanged should have minimal impact on D2S. However, we observe the following:
>
> | Method                  | SRCC $ \uparrow $   | PCC  $ \uparrow $  | RMSE  $ \downarrow $   | RMAE  $ \downarrow $   |
> | ----------------------- | ------ | ------ | ------ | ------ |
> | Baseline                | **0.9509** | **0.9544** | **0.0495** | **0.1962** |
> | Word-order shuffle 100% | 0.9479 | 0.9518 | 0.0517 | 0.2011 |
> | Word-order shuffle 50%  | 0.9508 | 0.9539 | 0.0512 | 0.1996 |
>
> The performance consistently drops when semantic structure in captions is disrupted ($ \approx 0.3–0.4 $ % SRCC/PLCC degradation). Notably, token frequencies remain nearly identical, so this degradation cannot be explained by distributional regularization alone. This indicates that the visual encoder does rely on meaningful semantic alignment with text during training.
>
> **(2) Linear probing on COCO-val**
>
> To further test whether semantic information is internalized by the visual encoder, we froze the D2S vision backbone and trained linear probes for COCO object detection ($ mAP_{50}=7.47 $ %). Compared with a vision-only model ($ mAP_{50}=6.12 $ %), D2S consistently improves detection AP(%) across a broad range of semantic categories:
>
> | COCO class  | bicycle  | motorcycle | airplane  | zebra    | giraffe  | skateboard | carrot   | ...  |
> | ----------- | -------- | ---------- | --------- | -------- | -------- | ---------- | -------- | ---- |
> | Vision-only | 3.18     | 2.87       | 13.45     | 3.39     | 3.10     | 2.58       | 4.15     | ...  |
> | D2S-R18     | **8.96** | **15.26**  | **18.11** | **5.76** | **5.26** | **5.32**   | **9.00** | ...  |
>
> The gains are substantial in semantically coherent classes, especially those with clear textual descriptions. If text supervision merely acted as random regularization, we would not expect the frozen D2S backbone to carry such improved semantic linear separability.
>
> Both results converge to the same conclusion. **(i)** When semantic structure is destroyed (word order shuffled), performance drops, indicating that D2S uses semantic relations rather than shallow token statistics. **(ii)** When the D2S backbone is probed on an external semantic task (COCO detection), it shows stronger semantic structure than a vision-only baseline.
>
> These two independent tests demonstrate that D2S does not rely on text solely as statistical regularization. The visual encoder internalizes transferable semantic information during training, and this semantic structure persists even after the text branch is removed at inference time. We will include these analyses and discussion in the revised version.
>
> > **Response to question 5**: Training-time overhead and performance trade-off of Entropy Alignment (EAL).
>
> We enlarge the buffer from 512 to 2048 entries gains +0.0055 SRCC / +0.0061 PCC at the cost of only $ \sim 1.4 \times $ training time; further increasing to 4096 entries yields negligible extra accuracy (+0.0002 SRCC) while time keeps growing $  \sim 2.4 \times $, confirming 2048 as the knee point. Hence the accuracy-time elbow occurs at 2048, which we adopt as the default. The extra 40% (*1.1 A2000 hour at 2048 entries*) training cost is acceptable for an offline task like ICA. Importantly, EAL introduces zero parameters and zero latency at inference, so the trade-off only exists during training. We will add the description of the training time in **Table 7** in the revised version.

---

> > ### Author Response · Authors · 2025-11-26
> > **Official Comment by Authors**
> >
> > Dear Reviewer QTaf,
> >
> > We hope this message finds you well. We wanted to follow up on the rebuttal we submitted over a week ago for our ICLR submission.
> >
> > We know the review period keeps you busy, and we really appreciate the time you've put into reviewing our paper. We've worked hard to address the concerns from your review. Whenever you have a moment, we'd love to hear what you think about our responses.
> >
> > If you need us to clarify anything else, just let us know. Thanks so much for your time.
> >
> > Best regards,
> >
> > Authors

---

> > > ### Comment · Reviewer_QTaf · 2025-11-26
> > >
> > > Dear Authors,
> > >
> > > Thank you for the effort you put into addressing my concerns. I am fully satisfied with your clarifications and will therefore support the acceptance of your manuscript.

---

> > > > ### Author Response · Authors · 2025-11-26
> > > > **Thank you for your acknowledgment!**
> > > >
> > > > Dear Reviewer QTaf,
> > > >
> > > > Thank you for your positive response to our rebuttal. We are delighted that our responses have addressed your concerns, and we truly appreciate your valuable feedback and encouragement.
> > > >
> > > > Best regards,
> > > >
> > > > The Authors of Submission 1036

---

### Author Response · Authors · 2025-11-19
**GENERAL RESPONSE (the summary of revisions)**

We would like to express our gratitude to Reviewers QTaf, xHZP, urPq, and WyHU for their diligent review and constructive feedback. Across the raised concerns, the core issues focus on **(1)** theoretical clarity of the proposed entropy and generalization analysis, **(2)** the necessity of semantic information and the role of captioning, **(3)** robustness of the D2S pipeline under variations in prompts, captioners, and alignment modules, and **(4)** the empirical significance of the proposed entropy distribution alignment (EAL) and feature alignment (FAL).
 We systematically addressed each point through additional explanations and expanded experiments. We have revised the manuscript (mark the modified content in **blue**), the details are as follows:

**Key Clarifications Provided.**

 • We clarified the assumptions behind **Proposition 1**, reworked the derivation in **Appendix A.1**, and made the entropy formulation and inequality stating conditions fully explicit.

 • We provided a clearer generalization argument via empirical Rademacher complexity and added textual explanations where needed.

 • We analyzed the necessity and contribution of high-level semantics, showing that semantics complement visual features but do not replace them; vision-only remains strong, while vision–text fusion gives consistent but moderate gains.

 • We provided detailed clarification on the role of the projection layer, the motivation of AttnPool, and how text guidance influences the visual encoder without being used at inference time.

**Additional Experiments Added.**

 • Caption robustness: shuffled captions, random sentences, meaningless tokens. (**Table 9**)

 • Alternative alignment modules: CLIP-style captions, CLIP-pretrained aligner, BLIP embeddings. (**Table 11**)

 • Prompt ablation matrix: single and multi-prompt combinations. (**Table 8**)

 • Projection layer ablation: with vs. without projection. (**Table 12**)

 • Full-data training on three external datasets: Savoias, Nagle4K, VISC-C. (**Table 6**)

 • Additional visualization:  channel utilization histogram and activation maps (comparison-based). (**Figure 7** and **Figure 16**)

$ \quad $

We thank the reviewers again for their insightful suggestions and look forward to further discussion as the review process continues.

---

### Author Response · Authors · 2025-11-28
**Gentle reminder**

Dear Reviewers,

We sincerely appreciate the time  you have dedicated to reviewing our paper.  As the discussion period is nearing its end, I would like to ensure that we have fully addressed all of your concerns. If there are any additional concerns or suggestions, please let us know. Your insights are invaluable to us, and we would be grateful for the opportunity to resolve any remaining issues and further improve our work.

Best regards,

The Authors of Submission 1036

---

### Author Response · Authors · 2025-11-30
**Summary of Contributions and Responses**

Dear Area Chair,

We hope this message finds you well. We sincerely thank you for accepting the assessment of our submission under these challenging circumstances. We recognize the additional time and effort required to evaluate the reverted discussion threads and highly appreciate your commitment to maintaining the integrity of the review process.

To assist your decision, we provide a brief structured summary in three parts below.

### **1. Main merits of the submission**

- **Training-time multimodal, inference-time unimodal ICA.**
   D2S uses captions and text–vision alignment (EAL/FAL) **only during training**, while **inference is purely image-based** with a standard ResNet backbone, injecting semantics without extra runtime cost compared to visual ICA baselines.
- **Complexity-aware theoretical framing.**
   The method is motivated by **entropy of fused multimodal representations** and **empirical Rademacher complexity**, leading to EAL/FAL as alignment mechanisms tailored specifically to scalar image complexity prediction rather than generic V+L fusion.
- **Strong, robust empirical behavior.**
   D2S reaches **state-of-the-art SRCC/PLCC on IC9600**, achieves **top performance on Savoias, Nagle4K, and VISC-C** with full training, and shows competitive NR-IQA transfer; additional analyses demonstrate robustness to captioner and prompt choices and provide qualitative visualizations of how text shifts visual attention to semantically meaningful regions.

### **2. Summary of rebuttal and revision**

The main concerns from QTaf, xHZP, urPq, and WyHU overlap substantially. The table below groups them into shared categories and summarizes how the revision answers each. We have uploaded the revisions and marked all the changes in **blue font**.

|Category|Raised by|Main concern (summary)|How the revision addresses it (condensed)|
|-|-|-|-|
|**Theory & Proposition 1**|QTaf, urPq, WyHU|Entropy definition; mapping features → distributions; validity of Eq.(2) for small $ \alpha,\beta $; operational support for “semantic compression / effective dimension”.| Make entropy definition explicit (softmax logits, $ -\sum p\log p $). Rephrase Proposition 1 as existence statement. Add regression check + synthetic test for Theorem 2. Add intrinsic-dimension analysis on trained model.|
|**Captions, prompts, BLIP dependence**|QTaf, xHZP, urPq, WyHU| Sensitivity to caption errors and template design; importance of the “overall complexity is …” prompt; dependence on BLIP vs other captioners/embeddings.|Prompt ablation matrix over P1–P4. Perturbation tests (shuffle, random, meaningless captions). Alternative captioners (Florence-2, Qwen3-VL-4B) + BLIP-embedding variant. Correlation analysis showing near-zero dependence on caption length/token counts.|
|**Projection module**|xHZP, WyHU|Projection is used only at training; risk of unintentionally biasing the visual encoder.|Ablation removing projection (use pooling + linear layer). SRCC/PLCC essentially unchanged, only slight RMSE/RMAE increase → projection is a light training-time adapter, not a critical dependency.|
|**Generalization beyond IC9600**|xHZP, WyHU, urPq|Results originally focused on IC9600; need full experiments on other ICA benchmarks.|Add full train/test on Savoias, Nagle4K, VISC-C. D2S-R50 attains top SRCC/PLCC on all three datasets among compared methods.|
|**Interpretability & role of EAL/FAL**|QTaf, WyHU| How text supervision modifies visual features; whether EAL/FAL do more than standard fusion/alignment blocks.|Add MaxMap/MeanMap/Grad-CAM comparisons vs vision-only, showing text shifts attention to semantic regions. Clarify EAL as entropy-distribution constraint and FAL as semantic-manifold anchoring specifically for complexity prediction.|

### **3. Reviewers siscussion status**

After the rebuttal and additional materials were posted:

- **Reviewer QTaf (score 8)** explicitly stated that his concerns had been resolved. His comments span theory, caption robustness, generalizability of EAL, semantic influence vs. pure regularization, training overhead, and qualitative analyses, all of which are covered by the revisions summarized above.
- **Reviewers xHZP, urPq, and WyHU (scores 4, 4, and 2)** did not post follow-up messages. Their concerns fall entirely into the five categories in Section 2 (theory, captions/prompts, projection, generalization, interpretability/EAL–FAL). Each category is now addressed by specific experiments and clarifications in the revised manuscript and supplemental materials.

Thank you once more for your diligent work in ensuring a fair review process. If you have any questions or require additional information, we are happy to provide further details.

Best regards,

The Authors of Submission 1036

---

### Note · Authors · 2026-01-26

I have read and agree with the venue's withdrawal policy on behalf of myself and my co-authors.

---

### Meta-Review · Area_Chair_dA19 · 2026-01-07

**Summary:**

The reviewers collectively raised substantial concerns regarding the novelty, experimental rigor, and clarity of the proposed method. While the idea of text-guided image complexity assessment is interesting, the paper fails to convincingly demonstrate its advantage over existing baselines or to establish the necessity of the proposed approach. Multiple reviewers questioned the validity of the experimental design, the limited evaluation datasets, and the unclear connection between the textual component and the final complexity metric. The rebuttal did not sufficiently resolve these fundamental issues.

**Reviewer Concerns:**

The rebuttal provided some clarifications regarding implementation details and additional experiments, which addressed minor concerns about reproducibility. However, the key issues remain unresolved: (1) the novelty remains marginal given prior work on multimodal complexity prediction; (2) the evaluation remains narrow and does not convincingly generalize beyond the chosen datasets; and (3) the claimed interpretability benefit of the textual guidance remains qualitative and unsubstantiated.

**Reviewer Scores:**

Based on the discussion, it is unlikely that any reviewer would significantly improve their scores. QTaf would keep the score as he/she claimed. urPq, xHZP and WyHU would likely maintain their low ratings (4, 4, 2) due to persisting concerns about novelty and empirical weakness.

---

### Decision · Program_Chairs · 2026-01-26

Reject